# Programmed cell senescence in skeleton during late puberty

Changjun Li[1,2], Yu Chai[1,3], Lei Wang[1,3], Bo Gao[1], Hao Chen[1], Peisong Gao [4], Feng-Quan Zhou[1], Xianghang Luo[2], Janet L. Crane[1,5], Bin Yu[3], Xu Cao[1] & Mei Wan[1]

Mesenchymal stem/progenitor cells (MSPCs) undergo rapid self-renewal and differentiation, contributing to fast skeletal growth during childhood and puberty. It remains unclear whether these cells change their properties during late puberty to young adulthood, when bone growth and accrual decelerate. Here we show that MSPCs in primary spongiosa of long bone in mice at late puberty undergo normal programmed senescence, characterized by loss of nestin expression. MSPC senescence is epigenetically controlled by the polycomb histone methyltransferase enhancer of zeste homolog 2 (Ezh2) and its trimethylation of histone H3 on Lysine 27 (H3K27me3) mark. Ezh2 maintains the repression of key cell senescence inducer genes through H3K27me3, and deletion of *Ezh2* in early pubertal mice results in premature cellular senescence, depleted MSPCs pool, and impaired osteogenesis as well as osteoporosis in later life. Our data reveals a programmed cell fate change in postnatal skeleton and unravels a regulatory mechanism underlying this phenomenon.

[1] Department of Orthopaedic Surgery, Johns Hopkins University School of Medicine, Baltimore, MD 21205, USA. [2] Department of Endocrinology, Endocrinology Research Center, The Xiangya Hospital of Central South University, Changsha, Hunan 410008, China. [3] Department of Orthopaedics and Traumatology, Nanfang Hospital, Southern Medical University, Guangzhou, Guangdong 510515, China. [4] Johns Hopkins Asthma & Allergy Center, Johns Hopkins University School of Medicine, Baltimore, MD 21224, USA. [5] Department of Pediatric Endocrinology, Johns Hopkins University School of Medicine, Baltimore, MD 21205, USA. Changjun Li and Yu Chai contributed equally to this work. Correspondence and requests for materials should be addressed to B.Y. (email: yubinol@163.com) or to M.W. (email: mwan4@jhmi.edu)

The skeleton is a remarkably adaptive organ, the development of which closely reflects the physiological stage. For example, skeletal growth is characterized by a sharp increase during early puberty, and deceleration and eventual cessation during late puberty[1,2]. As growth in length accelerates, bone mass accrual also increases markedly during childhood and adolescence until peak bone mass is achieved in early adulthood[3,4]. Elongation of long bones during the postnatal period and early puberty is driven primarily by chondrogenesis at the growth plates[5,6]. This process is followed by the co-invasion of blood vessels, osteoclasts, and mesenchymal stem/progenitor cells (MSPCs) that give rise to osteoblasts[7], leading to replacement of the cartilage template at the bottom of the growth plate by an ossified bony component, known as primary spongiosa[5]. In

**Fig. 1** Cellular senescence occurs in primary spongiosa of long bone during late puberty. **a–e** Representative senescence-associated β-galactosidase (SA-βGal) staining (blue) and quantitative analysis of SA-βGal+ cells in femoral metaphysis (**a–c**) and diaphysis (**d, e**) sections from increasing ages of male mice. 4, 6, 8, and 12W represent 4-, 6-, 8-, and 12-week-old mice, respectively. Images in **a** are lower power with boxes outlining the area of higher power in **b**. Numbers of SA-βGal+ cells per mm² tissue area in primary spongiosa (N. SA-βGal+ cells/PS.Ar) (**c**) and diaphysis (N. SA-βGal+ cells/DP.Ar) (**e**). Counterstained with eosin (pink). **f, g** Representative images of immunofluorescence staining (**f**) and quantitative analysis of ki67+ (**g**) cells (red) in femoral primary spongiosa from 4, 6, 8, and 12-week-old male mice. DAPI stains nuclei blue. Images in upper panels in **f** are lower power with boxes outlining the area of higher power in lower panels. Five mice per group. Data are represented as mean ± s.e.m. *Ar* tissue area, *DP* diaphysis, *GP* growth plate, *N* number, *PS* primary spongiosa. *P < 0.01 as determined by ANOVA

late puberty, the decline in growth rate is caused primarily by a decrease in the rate of chondrocyte proliferation in growth plate[8,9]. At this stage, cells at the primary spongiosa of long bone likely also undergo significant changes to adapt to the much slower bone growth/accrual in adulthood. Vascular endothelial cells that form invaded blood vessels and MSPCs that replenish

bone-forming osteoblasts are highly proliferative during bone growth, but these cells likely stop proliferating or are replaced by other cell types. It was reported that MSPCs isolated from the trabecular-rich metaphysis regions at two ends of a long bone have superior proliferative ability than the cells within the cortical-rich diaphysis[10]. However, little is known about change

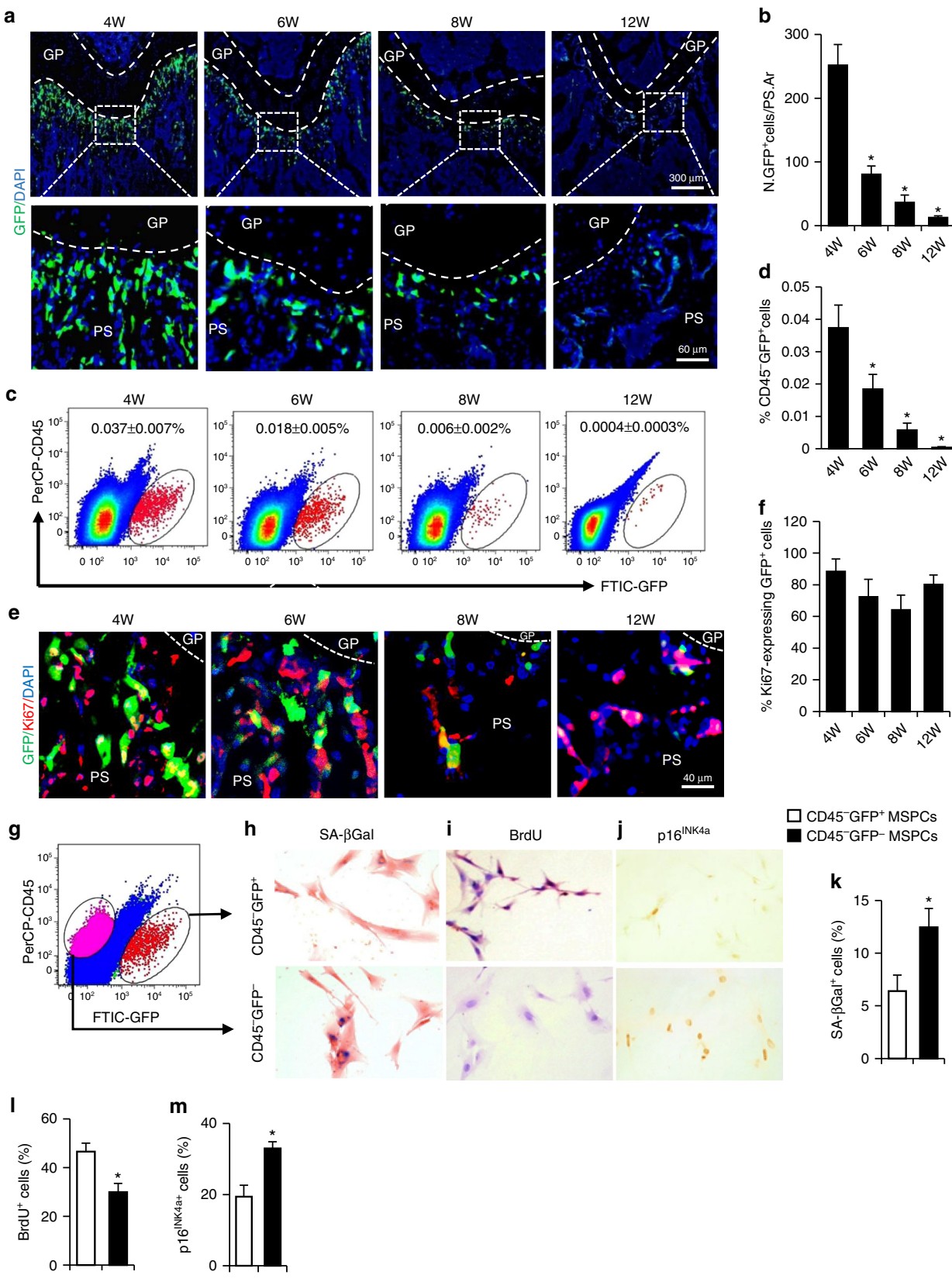

in the cells of primary spongiosa and the regulatory mechanisms in the skeleton during the transition from fast to slow growth.

Cellular senescence, a stable proliferative arrest that was implicated initially in aging and tumor suppression, can be induced by cellular damage or stress, including telomere attrition, DNA damage, activation of oncogenes, and oxidative stress[11,12]. These cells remain viable and metabolically active, but are refractory to mitogenic stimulation. Senescent cells exhibit essentially stable cell-cycle arrest through the actions of tumor suppressors such as p16[INK4a], p15[INK4b], p27[KIP1], retinoblastoma, p53, p21[CIP1], or others[13,14]. Other characteristics of senescent cells include increased lysosomal β-galactosidase activity (known as senescence-associated β-galactosidase or SA-βGal), senescence-associated secretory phenotype (SASP), and senescence-associated heterochromatin foci[12,15,16]. Recent studies suggest that cellular senescence not only contributes to organismal aging and aging-related diseases/disorders[13] but also plays an important role in embryonic development, tissue repair, wound healing, and protection against tissue fibrosis in physiologic conditions[17–20].

The concerted action of local niche signals and dynamic chromatin modifications reinforce stem cell fate decisions[21,22]. Upon changes in the local niche environment, stem/progenitor cells remodel chromatin to survive in transitional states, before undergoing fate selection. Several post-translational modifications of histones, including methylation, acetylation, phosphorylation and ubiquitination, lead to transcriptional regulation of gene expression in the cells. For example, the polycomb group (PcG) protein enhancer of zeste homolog 2 (Ezh2), the histone lysine demethylase Jmjd3, and the DNA methyltransferase Dnmt1 are important chromatin remodeling factors that regulate the activities of stem/progenitor cells[23,24]. Ezh2 is the functional enzymatic component of the polycomb repressive complex 2 (PRC2), which has histone methyltransferase activity and trimethylates primarily histone H3 on lysine 27 (i.e., H3K27me3), a mark of transcriptionally silent chromatin. Conversely, the methyl groups can be removed from H3K27 by histone demethylases Utx and Jmjd3, which demethylate H273K27me3 to H3K27me2 or H3K27me1[25]. Because of the essential role of the PRC2 complex in repressing many genes involved in somatic processes, the H3K27me3 mark is associated with the unique epigenetic state of stem/progenitor cells.

Given the beneficial role of cellular senescence in embryonic development, we asked whether senescence might also be involved in the cessation of bone growth/accrual during late puberty. We found that during late puberty, cells in primary spongiosa of long bone undergo senescence, which is also characterized by loss of expression of nestin, an intermediate filament protein. We also identify that the progression of cellular senescence is a normal programmed process governed by an epigenetic mechanism. Premature acquired senescence during early puberty leads to impaired angiogenesis and osteoblastogenesis as well as

bone loss in later adult life. Our data establishes MSPC senescence as a normal programmed cell fate change in postnatal skeleton and demonstrate proof-of-concept for targeting Ezh2-H3K27me3 in Juvenile osteoporosis.

## Results

**Cellular senescence occurs in long bone during late puberty.** Mouse pubertal growth can be divided into an early pubertal phase (3–5 weeks of age) and a late pubertal phase (5–8 weeks of age)[26]. We first assessed the rate of long bone growth in mice at different postnatal stages by measuring the femur lengths and calculating the elongation rates. Bone growth was fast during prepuberty and early puberty periods (2–5 weeks of age), became slow during late puberty (5–8 weeks of age), and almost stopped during young adulthood (8–12 weeks of age) in both female (Supplementary Fig. 1A, B) and male mice (Supplementary Fig. 1C, D). We then conducted exploratory SA-βGal staining in femoral bones of mice at a chosen age in each bone growth phase. A significant increase in the number of SA-βGal+ cells was observed only in primary spongiosa (i.e., trabecular bone adjacent to the growth plate) of the femur bones in 6- and 8-week-old male mice relative to 4-week-old mice, whereas the SA-βGal+ senescent cells diminished at the same region of femoral bone in 12-week-old male mice (Fig. 1a–c). SA-βGal+ cells were barely detectable in secondary spongiosa and the diaphyseal bone marrow in mice at all ages analyzed (Fig. 1d, e), indicating that senescence in pubertal bone is a time- and location-restricted event. Consistently, at 4 weeks (early pubertal phase), cells at the primary spongiosa were highly positive for the proliferation marker Ki67 (Fig. 1f, g). However, at 6 and 8 weeks (late pubertal phase), Ki67 staining was dramatically reduced in the same region. At 12 weeks, whereas SA-βGal+ senescent cells were almost undetectable, Ki67+ cells were restored in femoral primary spongiosa, indicating a strong association between SA-βGal activity and the absence of proliferation. Thus, cells in primary spongiosa of long bones undergo senescence-like growth arrest during late puberty when the bone growth rate starts to decline.

**Senescent MSPCs is featured by loss of nestin expression.** In adult bone, cells expressing GFP in response to a nestin promoter/enhancer (Nestin-GFP) and those expressing Nes-CreER are heterogeneous precursor cells mainly in endothelial and mesenchymal lineage[27–29]. We detected a progressive reduction of Nestin-GFP signaling in Nestin-GFP mice (Fig. 2a, b) and nestin+ cells in C57BL/6 mice (Supplementary Fig. 2A, B) specifically in femoral primary spongiosa in 6- and 8-week-old mice, coincident with the occurrence of senescent cells at the same region of long bone. We isolated mesenchymal stem/progenitor cells (MSPCs) from both bone marrow and endosteal bone surface in femoral metaphysis (including primary and secondary

**Fig. 2** Senescent MSPCs are characterized by loss of nestin expression. **a**, **b** Representative images of GFP immunofluorescence staining (green) and quantitative analysis of GFP+ cells in femoral primary spongiosa from 4, 6, 8, and 12-week-old male Nestin-GFP mice. Images in upper panels in **a** are lower power with boxes outlining the area of higher power in lower panels. Numbers of GFP+ cells per mm² tissue area in primary spongiosa (N. GFP+ cells/PS. Ar) **b**. 4W, 6W, 8W, and 12W represent 4-, 6-, 8-, and 12-week-old mice, respectively. DAPI stains nuclei blue. **c**, **d** Representative images of the flow cytometry analysis (**c**) and the percentage of the CD45⁻GFP+ cells (**d**) in femoral metaphysis from 4-, 6-, 8-, and 12-week-old male Nestin-GFP mice. **e**, **f** Double-immunofluorescence images of femoral metaphysis sections from 4-, 6-, 8-, and 12-week-old male Nestin-GFP mice using antibodies against GFP (green) and Ki67 (red) **e**. DAPI stains nuclei blue. GP growth plate. PS primary spongiosa. Quantification of the percentage of GFP+ cells that express Ki67 **f**. Five mice per group. Data are represented as mean ± s.e.m. *P < 0.05 as determined by ANOVA. **g** Diagram showing isolation of Nestin-GFP+ (red) and Nestin-GFP⁻ (purple) mesenchymal stem/progenitor cells (MSPCs) by fluorescence-activated cell sorting. Detailed information on the isolation of MSPCs from femoral metaphysis from Nestin-GFP mice are described in Supplementary Fig. 3 and Methods section. **h–j** The sorted cells were cultured, and the SA-βGal staining (**h**), BrdU incorporation (**i**), and p16[INK4a] immunostaining (**j**) were performed, and representative images were shown. **k–m** Quantification of the percentage of the cells that express SA-βGal (**k**), BrdU (**l**), and p16[INK4a] (**m**). n = 5. Data are represented as mean ± s.e.m. *P < 0.01 as determined by Student's t-tests

spongiosa) using an established enzymatic digestion approach[10] (Supplementary Fig. 3) because isolating cells from primary spongiosa has technical difficulty. Consistent with the immunofluorescence staining results, flow cytometry analysis of the isolated cells showed that the percentage of CD45−GFP+ cells was gradually reduced in mice of 6–12 weeks of age (Fig. 2c, d). We reasoned that cells expressing nestin may represent highly proliferative cells, and the cellular senescence in this region during late puberty may be the result of a gradual loss of nestin

expression. Indeed, double-immunofluorescence staining showed that most of Nestin-GFP+ cells (about 84%) expressed proliferation marker Ki67 (Fig. 2e, f), and only a small proportion of Nestin-GFP+ cells expressed senescence marker p16[INK4a] in primary spongiosa (Supplementary Fig. 2C, D). To further document the correlation of loss of nestin expression with cellular senescence, CD45−GFP+ and CD45−GFP− MSPCs were isolated, respectively, from the femoral metaphysis from 6-week-old mice as described in Supplementary Fig. 3 (Fig. 2g). CD45−GFP−

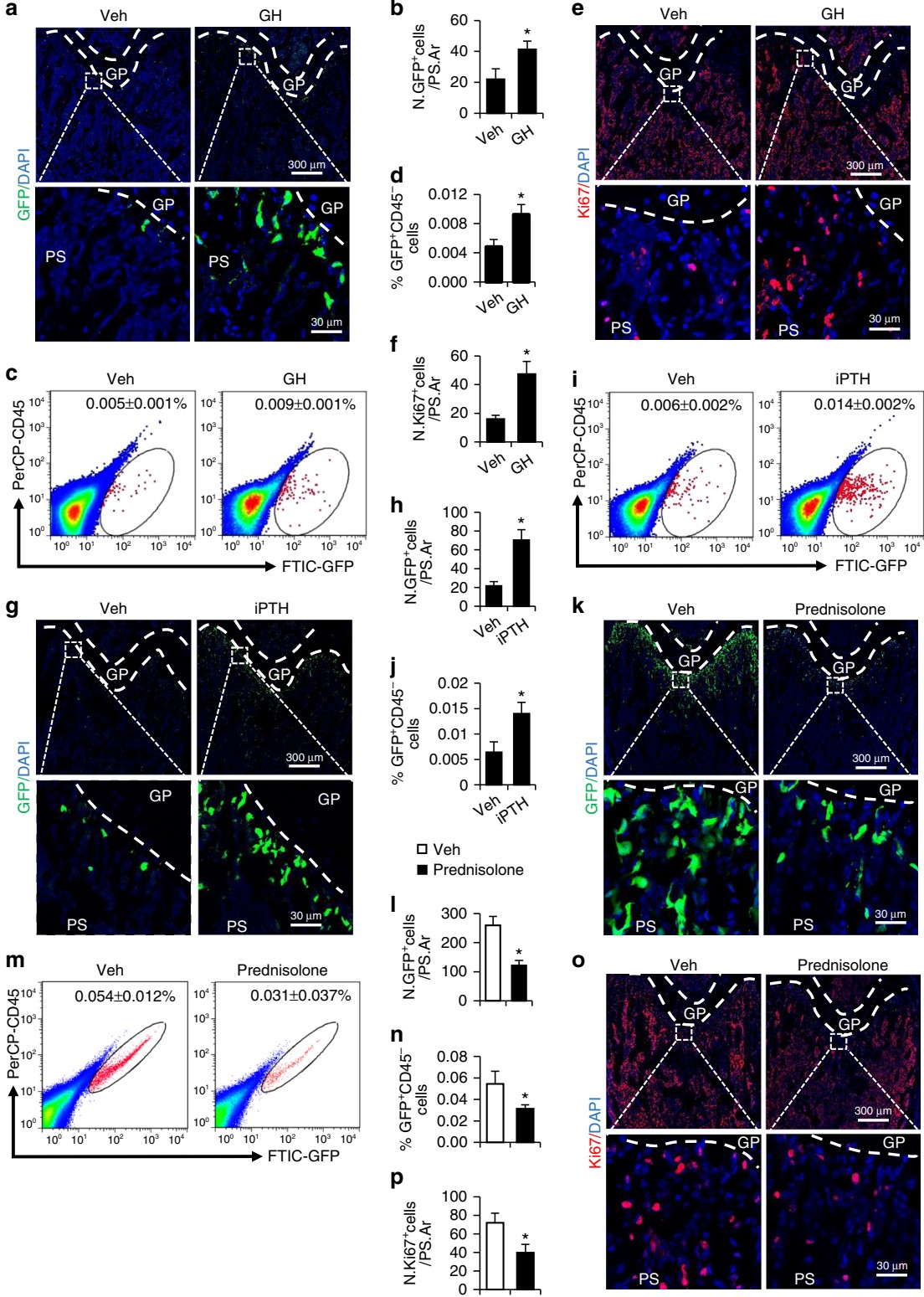

MSPCs, relative to CD45−GFP+ MSPCs, had increased SA-βGal+ cells (Fig. 2h, k), less BrdU labeling (Fig. 2i, l), and more cells positive for cellular senescence marker p16$^{INK4a}$ (Fig. 2j, m). Therefore, nestin expression maintains active proliferative property of the cells, whereas loss of nestin expression in MSPCs represents a distinctive feature for senescence-like cell cycle arrest in metaphysis region of long bone in late puberty. We found that most of GFP+ cells (about 82%) expressed osteoprogenitor marker osterix (Osx), and 11% of GFP+ cells expressed endothelial marker endomucin (Emcn) (Supplementary Fig. 4). Interestingly, leptin receptor (LepR)+ cells, multipotent MSPCs that give rise to most bone and adipocytes in adult bone marrow[30,31], exhibited an opposite expression pattern to nestin+ cells in femoral metaphysis. The number of LepR+ cells gradually increased in primary spongiosa in femora from 4 to 12 weeks of age, as detected by immunofluorescence staining (Supplementary Fig. 5A–C) and flow cytometry analysis (Supplementary Fig. 5D, E).

**Senescence of MSPCs is regulated by bone-regulatory agents.** We tested whether the progression of the cellular senescence in primary spongiosa of long bone, as indicated by loss of nestin expression, is changed by bone formation-regulating factors/ agents during childhood. Growth hormone (GH) is critical to longitudinal skeletal growth and bone accrual during the postnatal and pubertal periods[32], and an increase in bone mass during puberty is controlled largely by GH/insulin-like growth factor 1 (IGF-1) axis[33,34]. We assessed the changes in nestin expression in the femoral primary spongiosa in pubertal mice after administration of recombinant GH. Daily injection of GH into early pubertal mice (4 weeks old) for 4 weeks increased Nestin-GFP+ cells in primary spongiosa in late pubertal mice (8 weeks old) compared with vehicle-treated mice as detected by immunofluorescence staining (Fig. 3a, b). Similar results were obtained by flow cytometry analysis of the percentage of CD45−GFP+ cells in the metaphysis MSPCs (Fig. 3c, d). Consistently, the percentage of Ki67+ proliferative cells in the same region was higher in GH-treated mice relative to vehicle-treated mice (Fig. 3e, f). Intermittent PTH (iPTH) treatment stimulates bone accrual during childhood and puberty[35]. We also tested the change in the number of nestin+ cells in the femoral primary spongiosa in iPTH-treated pubertal mice. Significantly increased Nestin-GFP+ cells were observed in primary spongiosa of femora in 8-week-old late pubertal mice after 4 weeks of daily injection of PTH1-34 (Fig. 3g–j). Therefore, PTH inhibited the cell senescence in primary spongiosa of long bone in during puberty.

Long-term use of glucocorticoids impairs skeletal growth and bone accrual, leading to childhood osteoporosis and vertebral fractures[36]. We thus assessed whether glucocorticoid-treated early pubertal mice have accelerated cellular senescence in primary spongiosa by injecting mice daily with prednisolone during prepubertal and early pubertal period (from age 2 weeks to 4 weeks). The number of Nestin-GFP+ cells in femoral primary spongiosa was significantly reduced in prednisolone-treated mice relative to vehicle-treated mice as assessed by immunofluorescence staining and flow cytometry analysis (Fig. 3k–n). Prednisolone treatment also reduced Ki67+ proliferative cells (Fig. 3o, p) in the same region of the femur. Therefore, cellular senescence occurs prematurely in early pubertal mice after glucocorticoid treatment.

**Ezh2 represses senescence inducer genes in MSPCs by H3K27me3.** To examine whether spatiotemporal-restricted cellular senescence is associated with epigenetic changes, we isolated CD45−GFP+ and CD45−GFP− MSPCs from the femoral metaphysis of 6-week-old mice as described in Supplementary Fig. 3. Profiling the expression of 86 key genes encoding enzymes that modify genomic DNA and histones showed 7 differentially expressed genes in these two cell populations (1 downregulated and 6 upregulated; Fig. 4a). Among these genes, Ezh2, a histone methyltransferase within the PcG protein complex and that specifically catalyzes H3K27me3, was the only enzyme showing significant downregulation in CD45−GFP− MSPCs relative to CD45−GFP+ MSPCs. Consistently, further qRT-PCR analysis showed more than four-fold reduction in the expression of Ezh2 in CD45−GFP− MSPCs compared with that in CD45−GFP+ MSPCs (Fig. 4b). On the contrary, the expression of Ezh1, another histone methyltransferase within PcG protein complex that also targets H3K27, increased in CD45−GFP− MSPCs versus CD45−GFP+ MSPCs, indicating that elevated Ezh1 may play a compensatory role in preventing the complete loss of H3K27me3 in the cells.

To examine whether Ezh2 regulates cellular senescence by directly modulating the trimethylation of histone H3 lysine 27 (H3K27me3), we performed chromatin immunoprecipitation (ChIP)-qPCR assays to assess changes in histone methylation status at the promoter regions of the key genes involved in cell senescence and cell cycle arrest. Several studies demonstrated that Ezh2 increases the repressive mark H3K27me3 at INK4b-ARF-INK4a locus (Fig. 4c), which encodes INK4 family of cyclin-dependent kinase inhibitors, p15$^{INK4b}$ and p16$^{INK4a}$, and a tumor suppressor p19/p14$^{ARF}$[37–39]. We therefore examined the status of H3K27me3 along the INK4b-ARF-INK4a locus in CD45−GFP+ and CD45−GFP− MSPCs using ChIP assays. Significantly enriched H3K27me3 was detected in the regions just upstream of the transcription start site (TSS) (p16-1), surrounding the TSS (p16-2), and within the intron just downstream of TSS (p16-3) of p16$^{INK4a}$ in the Nestin-GFP+ MSPCs, but not in Nestin-GFP− MSPCs isolated from the femoral metaphysis (Fig. 4d). Similarly,

**Fig. 3** Senescence of MSPCs is regulated by bone formation-regulatory agents. **a–f** Four-week-old male *Nestin-GFP* mice were treated with recombinant mouse-GH (5 mg/kg B.W.) or vehicle (Veh) by daily intraperitoneal injection for 4 weeks. Immunofluorescence staining of femur sections were performed using antibodies against GFP (green) (**a**) or Ki67 (red) (**e**). Images in upper panels in **a** and **e** are lower power with boxes outlining the area of higher power in lower panels. Quantification of the number of GFP+ (**b**) or Ki67+ (**f**) cells in femoral primary spongiosa. Representative images of flow cytometry analysis (**c**) and the quantification of the percentage of the CD45−GFP+ cells isolated from femoral metaphysis (**d**). n = 5. Data are represented as mean ± s.e.m. *P < 0.01 as determined by Student's t-tests. **g–j** Four-week-old male *Nestin-GFP* mice were treated with human PTH1-34 (80 μg/kg B.W.) or Veh by daily subcutaneous injection for 4 weeks. Immunofluorescence staining of femur sections were performed using antibodies against GFP (green) (**g**). Images in upper panels in **g** are lower power with boxes outlining the area of higher power in lower panels. Quantification of the number of GFP+ (**h**) cells in femoral primary spongiosa. Representative images of flow cytometry analysis (**i**) and the quantification of the percentage of the CD45−GFP+ cells isolated from femoral metaphysis (**j**). n = 5. Data are represented as mean ± s.e.m. *P < 0.01 as determined by Student's t-tests. **k–p** Two-week-old male *Nestin-GFP* mice were treated with prednisolone (10 mg/m²/day) or Veh by daily intraperitoneal injection for 2 weeks. Immunofluorescence staining of femur sections were performed using antibodies against GFP (green) (**k**) or Ki67 (red) (**o**). Images in upper panels in **k**, **o** are lower power with boxes outlining the area of higher power in lower panels. Quantification of the number of GFP+ (**l**) or Ki67+ (**p**) cells in femoral primary spongiosa. Representative images of flow cytometry analysis (**m**) and the quantification of the percentage of the CD45−GFP+ cells isolated from femoral metaphysis (**n**). n = 5. Data are represented as mean ± s.e.m. *P < 0.01 as determined by Student's t-tests. GP growth plate, PS primary spongiosa

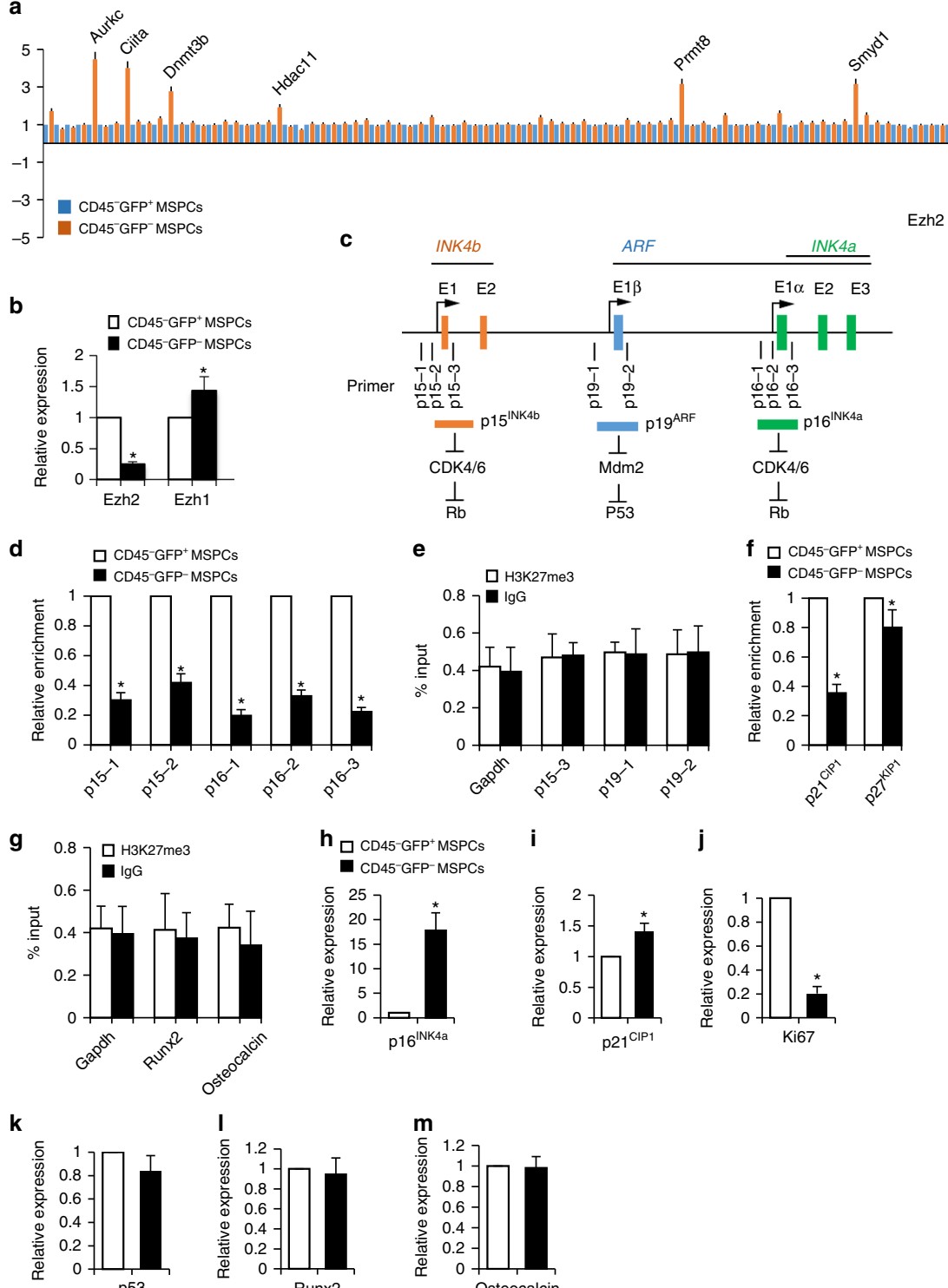

**Fig. 4** Ezh2 represses cell senescence inducer genes in MSPCs via H3K27me3. **a** CD45⁻GFP⁺ and CD45⁻GFP⁻ mesenchymal stem/progenitor cells (MSPCs) were separately isolated from femoral metaphyses from 4-week-old male *Nestin-GFP* mice, and the total RNA was subjected for PCR array analysis of the chromatin modification enzymes. **b** Validation of messenger RNA changes by qRT-PCR analysis. **c–g** Chromatin immunoprecipitation (ChIP)-qPCR assays with H3K27me3 antibody or IgG antibody were performed using CD45⁻GFP⁺ and CD45⁻GFP⁻ MSPCs. Schematic representation of the *INK4a-ARF-INK4b* locus in mouse **c**. Rectangles indicate coding exons of the three genes (*p15^INK4b*, *p19^ARF* and *p16^INK4a*) separated by intronic sequences (black horizontal line). The regions chosen for PCR amplification by the primers in ChIP-qPCR assays are also indicated. ChIP and input DNA were measured using RT-qPCR with specific primers targeting the promoter regions of senescence-associated genes (**d–f**). ChIP and input DNA were measured using quantitative RT-qPCR with specific primers targeting the promoter regions of osteogenic genes (**g**). **h–m** Quantitative RT-PCR analysis of *p16^INK4a* (**h**), *p21^CIP1* (**i**), *ki67* (**j**), *p53* (**k**), *Runx2* (**l**), *Osteocalcin* (**m**) expression in the sorted CD45⁻GFP⁺ and CD45⁻GFP⁻ MSPCs. $n = 3$, Data are represented as mean ± s.e.m. *$P < 0.01$ as determined by Student's *t*-tests

H3K27me3 mark was also enriched in the regions just upstream of TSS (p15-1) and surrounding the TSS (p15-2) of $p15^{INK4b}$ in the Nestin-GFP$^+$ MSPCs isolated from the femoral metaphysis and reduced in Nestin-GFP$^-$ MSPCs (Fig. 4d). There was no specific H3K27me3 detected at the intron region near TSS (p15-3) of $p15^{INK4b}$, the regions just upstream of TSS (p19-1) and

surrounding the TSS (p19-2) of $p19^{ARF}$, as well as the TSS region of *Gapdh* (negative control) compared to the IgG control antibody in the cells (Fig. 4e). Furthermore, the promoter regions of cell cycle inhibitor genes $p21^{CIP1}$ and $p27^{KIP1}$ were also enriched in H3K27me3 in the Nestin-GFP$^+$ MSPCs isolated from the femoral metaphysis whereas this histone modification on the

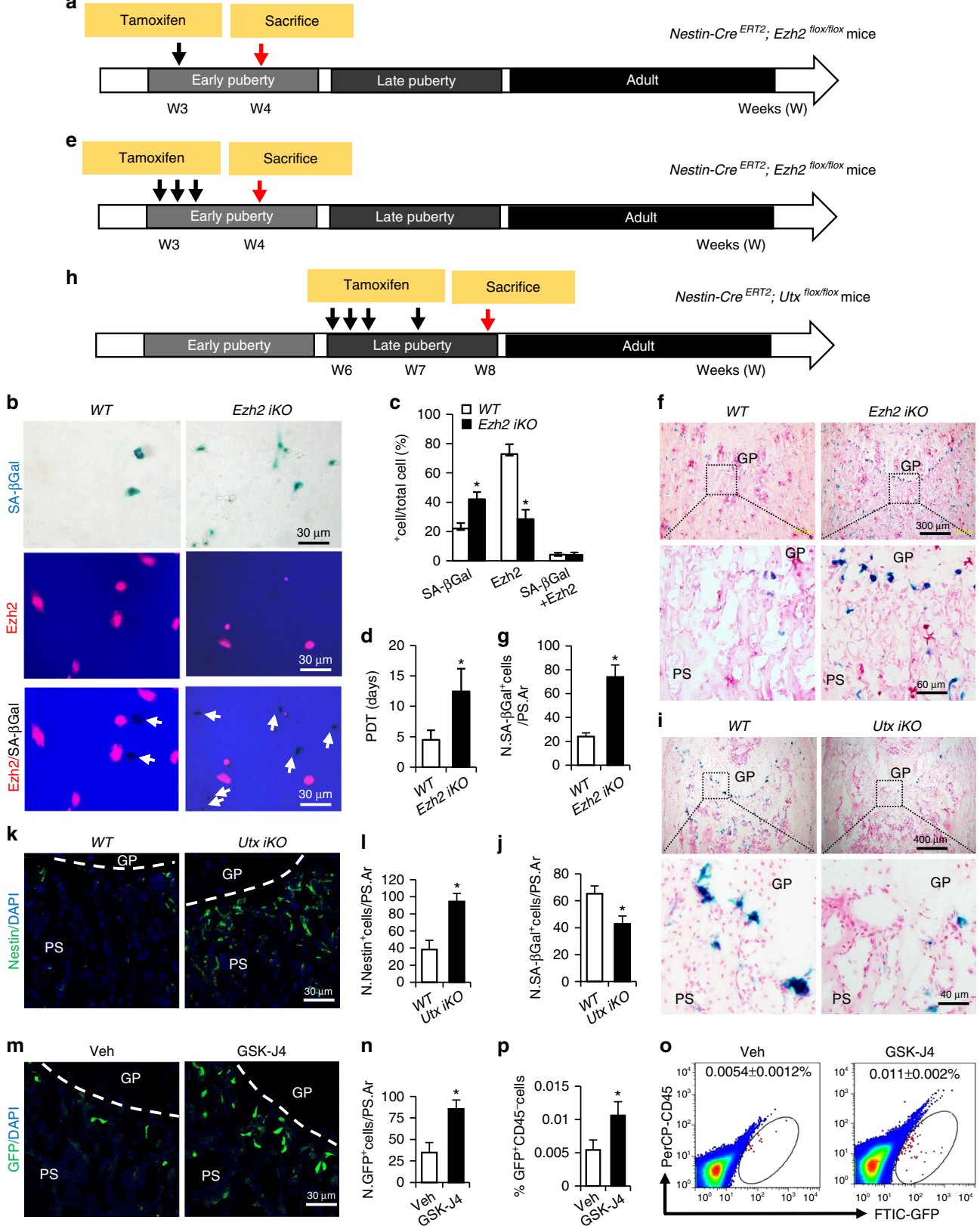

two genes was decreased in Nestin-GFP⁻ MSPCs (Fig. 4f). However, no specific H3K27me3 was detected at the TSS regions of master osteoblast differentiation inducer genes *Runx2* and *Osteocalcin* in the cells (Fig. 4g).

To further establish the relationship between the H3K27me3 mark change at cell senescence/cell cycle inhibitor genes and the change of their expression levels in MSPCs, we performed quantitative RT-PCR (qRT-PCR) and compared the expression of the genes in the populations of CD45⁻GFP⁻ and CD45⁻GFP⁺ MSPCs. Both the levels of $p16^{INK4a}$ (Fig. 4h) and $p21^{CIP1}$ (Fig. 4i) were significantly increased in CD45⁻GFP⁻ MSPCs compared with those in CD45⁻GFP⁺ MSPCs. Intriguingly, more than fivefold decrease in Ki67 expression were detected in the CD45⁻GFP⁻ MSPCs vs. the CD45⁻GFP⁺ MSPCs (Fig. 4j). No difference of the expression of *p53*, a downstream target of $p19^{ARF}$ that controls the senescence response to tissue damage or cancer-causing stress, was seen in these two cell populations (Fig. 4k). This data together with the undetected H3K27me3 mark at the TSS of the $p19^{ARF}$ promoter suggests that $p19^{ARF}$–*p53* pathway is not involved in this process. Consistent with the ChIP-qPCR data in Fig. 4g, we did not detect changes in the expression levels of *Runx2* and *Osteocalcin* in the CD45⁻GFP⁻ MSPCs vs. the CD45⁻GFP⁺ MSPCs (Fig. 4l, m). Thus, Ezh2-H3K27me3 maintains the repression of senescence inducer genes, and loss of the epigenetic mark leads to activation of these genes for cellular senescence.

**Progression of senescence is controlled by Ezh2-H3K27me3.** To further examine whether Ezh2-H3K27me3 is required for maintaining the nestin⁺ cells in primary spongiosa of long bone during early puberty, we generated an inducible *Nestin-Cre^ERT2::Ezh2^flox/flox* (*Ezh2 iKO*) mouse model by crossing *Ezh2^flox/flox* mice with *Nestin-Cre^ERT2* mice. In *Ezh2 iKO* mice, *Ezh2* was deleted specifically in nestin⁺ cells in a Tamoxifen-dependent manner. We first assessed whether deletion of *Ezh2* in nestin⁺ cells during early puberty causes senescence of these cells in primary spongiosa of long bones by administering a single dose of tamoxifen in 3-week-old *Nestin-Cre^ERT2* (*WT*) and *Ezh2 iKO* mice and isolated the MSPCs from metaphysis region of femoral bone 1 week later (Fig. 5a). Although most cells isolated from *WT* mice were Ezh2-positive (>75%), fewer than 30% of the cells from the *Ezh2 iKO* mice expressed *Ezh2* (Fig. 5b, c). The results suggest that *Ezh2* was deleted in more than half of the MSPCs at metaphyseal bone with one dose of Tamoxifen. The percentage of SA-βGal⁺ cells in the *Ezh2 iKO* mice was more than double that in *WT* mice (Fig. 5b, c). The expressions of *Ezh2* and SA-βGal were mutually exclusive in the cells, indicating a critical role of Ezh2 in preventing cellular senescence. The population doubling time of the

cells isolated from *Ezh2 iKO* mice was longer compared with those from *WT* mice, indicating slower growth rate (Fig. 5d). To further document the role of Ezh2-modulated H3K27me3 in protecting the cells from senescence, we injected 3 doses of tamoxifen (every other day) into 3-week-old *WT* and *Ezh2 iKO* mice (Fig. 5e). Significantly elevated SA-βGal⁺ senescent cells were detected in femoral primary spongiosa of the *Ezh2 iKO* mice compared with those in the *WT* mice (Fig. 5f, g).

The X chromosome-encoded histone demethylase Utx (also known as Kdm6a) mediates removal of repressive H3K27me3 to establish transcriptionally permissive chromatin. To examine whether the increased H3K27me3 level in nestin⁺ cells is sufficient to decelerate the senescence of nestin⁺ cells in primary spongiosa of long bone during late puberty, we employed an inducible KO mouse that specifically ablates *Utx* in nestin⁺ cells, i.e., *Nestin-Cre^ERT2;Utx^flox/flox* mice (*Utx iKO*). *Uty* is the Y-chromosome homolog of *Utx* and has overlapping redundancy with *Utx* in embryonic development. To eliminate the potential for confounding effects of Uty function in these experiments, we used female mice to analyze cellular senescence. Six-week-old female *Utx iKO* mice and *Nestin-Cre^ERT2* (*WT*) mice were injected with tamoxifen for 2 weeks (three doses in the first week, and one dose in the second week) (Fig. 5h). Significantly reduced SA-βGal⁺ senescent cells were detected in femoral primary spongiosa of the *Utx iKO* mice compared with those in the *WT* mice (Fig. 5i, j). Furthermore, increased number of nestin⁺ cells in primary spongiosa were detected in the *Utx iKO* mice (Fig. 5k, l). The number of nestin⁺ cells also significantly increased in femoral primary spongiosa of mice after a systemic administration of GSK-J4[40], a selective inhibitor of Jmjd3 and Utx (Fig. 5m–p). The results suggest that Ezh2-mediated H3K27me3 is required to maintain highly proliferative nestin⁺ cells in primary spongiosa of long bone in fast-growing bone.

**Deletion of *Ezh2* in nestin⁺ cells impairs osteogenesis.** We assessed whether *Ezh2* deletion in nestin⁺ cells during early puberty causes alterations of blood vessels and osteogenesis in the primary spongiosa of femora during late puberty. After three doses of tamoxifen administration (every other day) into 3-week-old *WT* and *Ezh2 iKO* mice, we assessed the bone phenotype in mice at 4 weeks of age (Fig. 6a). Although nestin⁺ cells were abundant in primary spongiosa of femoral bone in *WT* mice, these cells significantly decreased in *Ezh2 iKO* mice (Fig. 6b, c). CD31^high Emcn^high blood vessels in primary spongiosa, which are considered to be osteogenesis-coupled vessels[41–43], were also significantly reduced in the *Ezh2 iKO* mice relative to *WT* mice (Fig. 6d, e). A similar reduction in the Osx⁺ osteoprogenitor cells (Fig. 6f, g) and bone surface osteocalcin⁺ osteoblasts (Fig. 6h, i) in

**Fig. 5** Ezh2-H3K27me3 regulates progression of the senescence of MSPCs. Schematic diagram indicating the experimental workflow in different genetic *Ezh2* (**a**, **e**) or *Utx* (**h**) ablation mouse models. **a–d** A single dose of tamoxifen (100 mg/kg B.W.) was injected in 3-week-old male *Nestin-Cre^ERT2* (*WT*) and *Ezh2 iKO* mice and isolated the MSPCs from metaphysis region of femoral bone 1 week later. Co-staining of the SA-βGal and Ezh2 expression indicates that SA-βGal and Ezh2 are mutually expressed in cells (**b**). White arrows represent SA-βGal⁺ cells in merged images. The percentage of the cells that express SA-βGal, Ezh2, or both (**c**). Population doubling time (PDT) of the cells was measured (**d**). **f, g** Three-week-old male *Nestin-Cre^ERT2::Ezh2^flox/flox* (*Ezh2 iKO*) mice and *Nestin-Cre^ERT2* mice (*WT*) were injected with three doses of tamoxifen (100 mg/kg B.W., every other day). Representative SA-βGal staining and quantitative analysis of SA-βGal⁺ cells in femoral primary spongiosa (**f**). Numbers of the SA-βGal⁺ cells per mm² tissue area in primary spongiosa (N. SA-βgal⁺ cells/PS.Ar) (**g**). **i–l** Six-week-old female *Nestin-Cre^ERT2::Utx^flox/flox* (*Utx iKO*) mice and *Nestin-Cre^ERT2* mice (*WT*) were injected with tamoxifen for 2 weeks (100 mg/kg B.W., three doses during the first week, and one dose during the second week). Representative SA-βGal staining (**i**) and quantitative analysis of SA-βGal⁺ cells (**j**) in femoral primary spongiosa. Number of SA-βGal⁺ cells per mm² tissue area in primary spongiosa (N. SA-βgal⁺ cells/PS.Ar). Representative immunofluorescence staining using antibodies against nestin (**k**) and quantitative analysis of nestin⁺ cells (**l**) in femoral primary spongiosa. Number of nestin⁺ cells per mm² tissue area in primary spongiosa (N. nestin⁺ cells/PS.Ar). **m–p** Six-week-old male *Nestin-GFP* mice were treated with GSK-J4 (100 mg/kg B.W.) or vehicle by daily intraperitoneal injection for 2 weeks. Immunofluorescence staining of femur sections were performed using antibodies against GFP (green) (**m**). Quantification of the number of GFP⁺ (**n**) cells in femoral primary spongiosa. Representative images of flow cytometry analysis (**o**) and the quantification of the percentage of the CD45⁻GFP⁺ cells isolated from femoral metaphysis (**p**). n = 5. Data are represented as mean ± s.e.m. *GP* growth plate. *PS* primary spongiosa. *P < 0.01 as determined by Student's *t*-tests

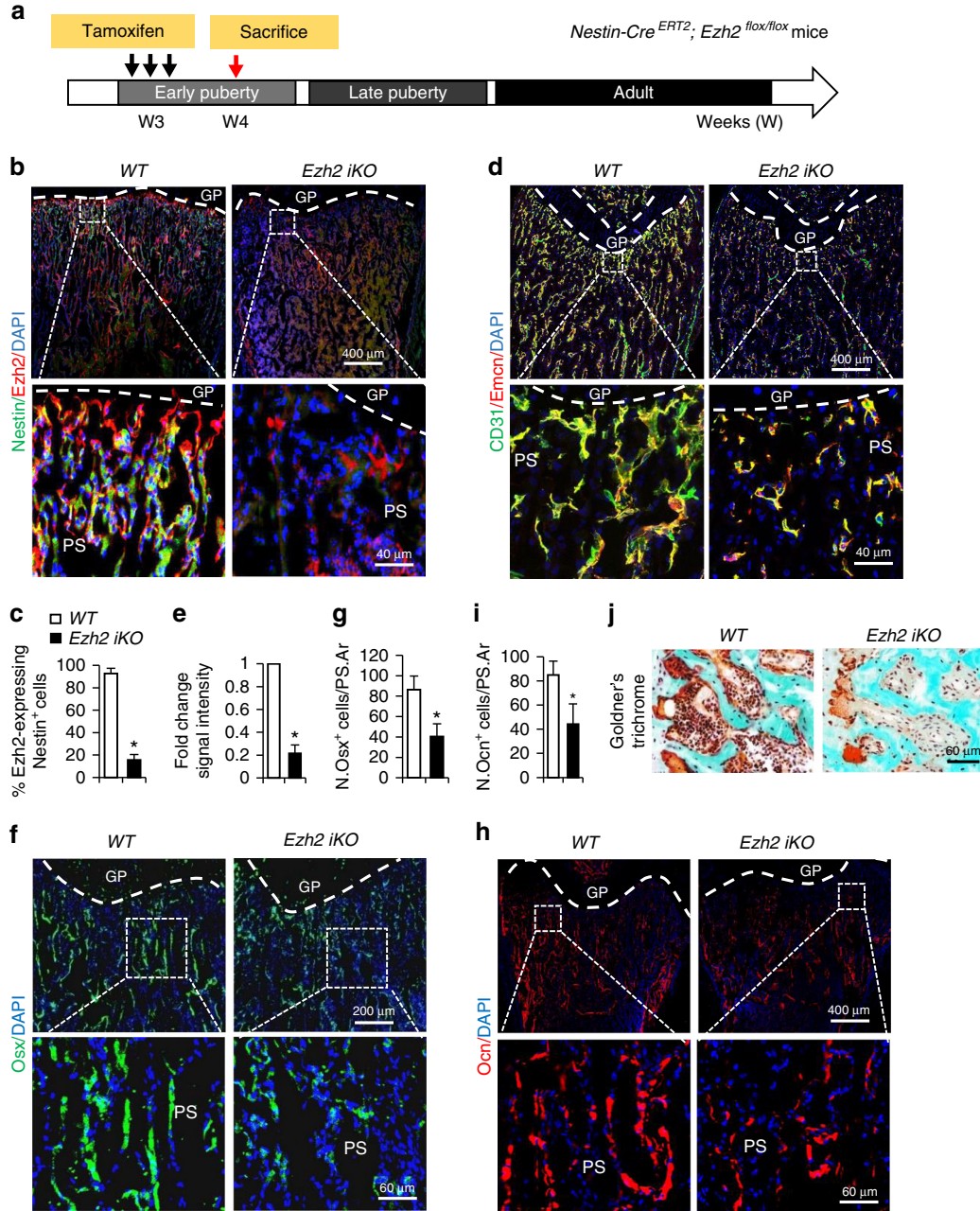

**Fig. 6** Deletion of *Ezh2* in nestin[+] cells during early puberty impairs osteogenesis. **a** Schematic diagram indicating the experimental workflow in *Nestin-Cre^ERT2::Ezh2 flox/flox* (*Ezh2 iKO*) mice. In short, three-week-old male *Ezh2 iKO* mice and *Nestin-Cre^ERT2* mice (*WT*) were injected with three doses of tamoxifen (100 mg/kg B.W., every other day. The mice were humanely killed at 4 weeks of age. Double-immunofluorescence staining of femoral metaphysis sections was performed using antibodies against nestin (green) and Ezh2 (red) **b**. Quantification of the percentage of nestin[+] cells that express Ezh2 in primary spongiosa (**c**). Double-immunofluorescence staining of femur sections was performed using antibodies against CD31 (green) and endomucin (Emcn) (red) (**d**). Quantitative analysis of relative fluorescence intensities in primary spongiosa (**e**). Immunofluorescence staining of femur sections using antibodies against osterix (Osx) (**f**) and osteocalcin (Ocn) (**h**). DAPI stains nuclei blue. Quantitative analysis of Osx[+] and Ocn[+] cells in primary spongiosa is shown in **g**, **i**, respectively. Representative images of trichrome staining of the metaphyseal trabecular bone (**j**). Five mice per group. Data are represented as mean ± s.e.m. *$P < 0.05$ as determined by Student's *t*-tests

the same region were also observed in the iKO mice. We examined whether osteoblastic new bone formation was affected by deletion of *Ezh2* in nestin[+] cells. Goldner's Trichrome staining showed significantly decreased newly formed bone in the primary spongiosa region in the *Ezh2 iKO* mice compared with the *WT* mice (Fig. 6j). Therefore, Ezh2 downregulation or deficiency during early puberty leads to premature senescence of the cells and subsequent loss of blood vessel formation and new bone formation.

**Loss of *Ezh2* in childhood causes low bone mass in adult life**. Finally, we tested whether the reduced osteoblastic bone formation in mice induced by Ezh2 deficiency during early puberty results in bone deficit during adulthood. We administered three doses of tamoxifen to 3-week-old *WT* and *Ezh2 iKO* mice, and the mice were killed at 16 weeks of age (Fig. 7a). Body weight and femoral length were not decreased in male or female *Ezh2 iKO* mice compared with *WT* controls at 16 weeks of age (Supplementary Fig. 6A–D). Femora were

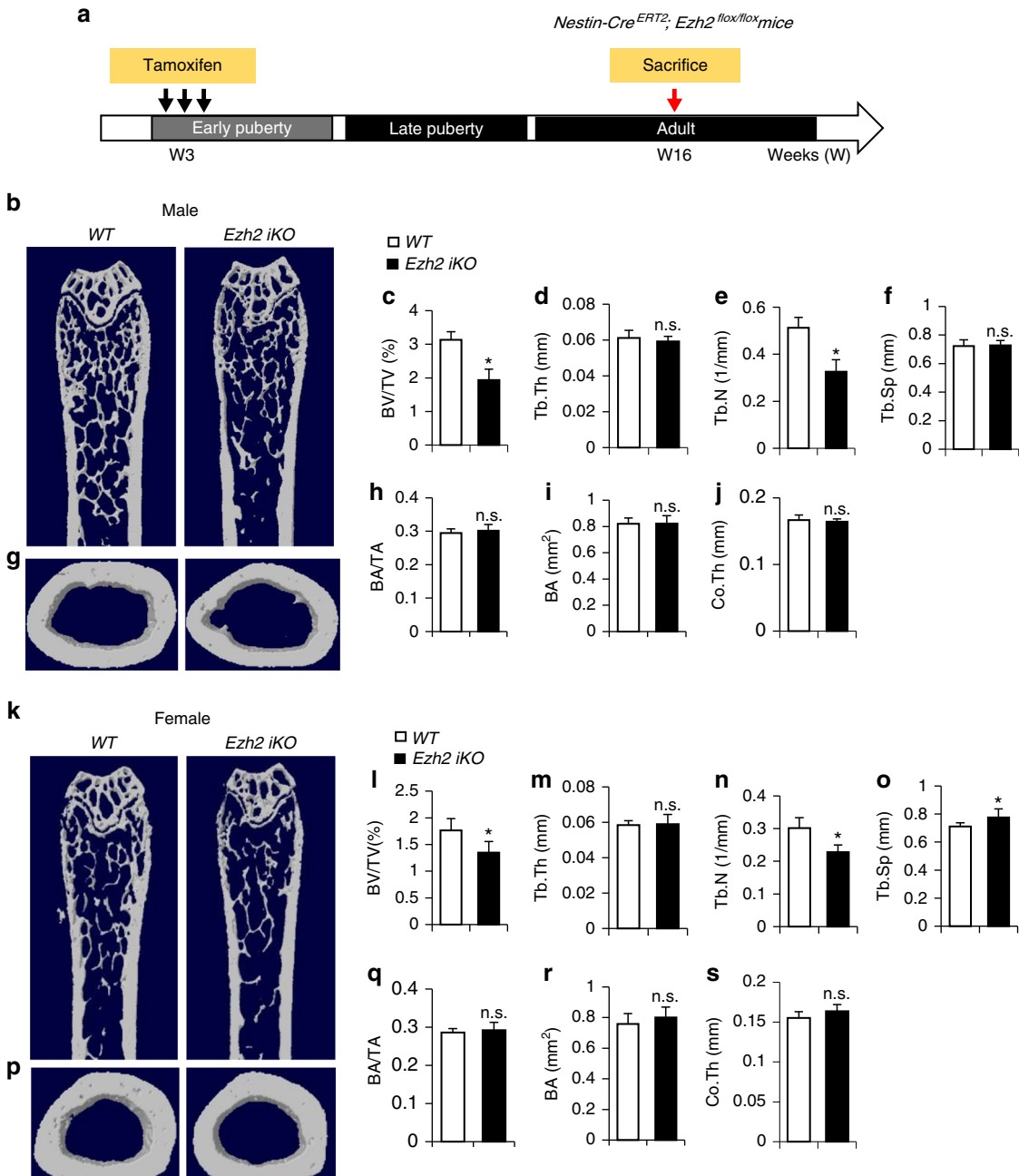

**Fig. 7** Deletion of *Ezh2* in nestin[+] cells in early puberty results in bone loss in later life. **a** Schematic diagram indicating the experimental workflow in *Nestin-Cre*[ERT2]*::Ezh2* [flox/flox] (*Ezh2 iKO*) mice. In short, three-week-old male and female *Ezh2 iKO* mice and *Nestin-Cre*[ERT2] mice (*WT*) were injected with 3 doses of tamoxifen (100 mg/kg B.W., every other day). The mice were humanely killed at 16 weeks of age. Representative µCT images of distal femur in male mice were shown in **b**. Quantitative analyses of Trabecular bone volume fraction (BV/TV) (**c**), trabecular thickness (Tb.Th) (**d**), trabecular number (Tb.N) (**e**), and trabecular separation (Tb.Sp) (**f**). Representative µCT images of cross-sections of femoral mid-diaphysis (**g**). The ratio of cortical bone area per total area (BA/TA) (**h**), cortical bone area (BA) (**i**), and cortical bone thickness (Co.Th) (**j**). Representative µCT images of distal femur in female mice were shown in **k**. Quantitative analyses of Trabecular bone volume fraction (BV/TV) (**l**), trabecular thickness (Tb.Th) (**m**), trabecular number (Tb.N) (**n**), and trabecular separation (Tb.Sp) (**o**). Representative µCT images of cross-sections of femoral mid-diaphysis (**p**). The ratio of cortical bone area per total area (BA/TA) (**q**), cortical bone area (BA) (**r**), and cortical bone thickness (Co.Th) (**s**). Ten mice per group; *GP* growth plate. *PS* primary spongiosa. Data are represented as mean ± s.e.m. *$P < 0.05$ as determined by Student's *t*-tests

harvested from 16-week-mice, and bone architecture of the femoral bone was measured by µCT. A significant reduction in the mass of trabecular bone was observed in both male and female *Ezh2 iKO* mice relative to sex-matched *WT* mice (Fig. 7b, k). The *Ezh2 iKO* mice exhibited reduced trabecular bone volume and number and greater trabecular bone separation compared with *WT* littermates in male (Fig. 7c–f) and female mice

(Fig. 7l–o). No significant differences were observed for the cortical bone parameters including bone area, bone area/total area, and cortical bone thickness in the *Ezh2 iKO* mice as compared to their *WT* littermates in male (Fig. 7g–j) and female mice (Fig. 7p–s). Therefore, premature cellular senescence in primary spongiosa during the prepubertal or early pubertal phase leads to osteoporosis in later life.

## Discussion

Here we report that MSPCs in the primary spongiosa of long bones undergo cellular senescence during late puberty when the speed of bone growth/accrual starts to slow. The senescence process is programed by a conserved mechanism because it restricts in a particular region of long bone and follows a specific time course. Cellular senescence was defined by the presence of a senescence marker SA-βGal and a key senescence mediator p16$^{INK4a}$ detected in the bone tissue sections. Cell senescence was also supported by a reduced proliferation rate, as detected by BrdU labeling and the expression of proliferation marker Ki67. Metaphyses, especially the primary spongiosa that lies immediately adjacent to the growth plate, are the zone of active neoangiogenesis and osteogenesis during the rapid postnatal and pubertal skeletal growth periods. During long bone growth, the whole metaphysis is modeled. We found that cell senescence was barely detectable during early puberty, became obvious during late puberty, and was much reduced in adulthood in the primary spongiosa of mouse femora. There were many fewer senescent cells in the secondary spongiosa relative to the primary spongiosa, and senescent cells were almost undetectable in the diaphysis in the center bone marrow area, indicating that the cellular senescence may represent the primary spongiosa involuting during the late puberty. We also found that the senescence of the cells was reduced in mice treated with bone anabolic agents GH and iPTH but accelerated in mice treated with prednisolone, indicating that the progression of this process can be manipulated by external factors known to affect skeletal growth and bone accrual. Cellular senescence in primary spongiosa is an important signature for the transition from fast to slow growing phase in long bones.

We identify that the gradual loss of nestin expression is another feature of cellular senescence in the primary spongiosa of long bones during late puberty. Nestin, a type IV intermediate filament protein expressed in many progenitor cell types[44,45], is required for self-renewal, proliferation and cell cycle progression of the cells[46,47]. It was previously noted that there is a loss of nestin-expressing cells in primary spongiosa of long bones as mice grow into adults[27]. However, the physiologic interpretation of this phenomenon are unclear. Given that other intermediate filament proteins are known to be expressed when a cell is dividing, it is logical to postulate that nestin is the preferred intermediate filament for cell division of bone marrow MSPCs. We identified that most nestin-expressing cells at the primary spongiosa also expressed the proliferation marker Ki67 and were labeled by BrdU, whereas the cellular senescence marker p16$^{INK4a}$ was exclusively expressed in the Nestin-GFP-negative cell population. An earlier study showed that MSPCs residing in metaphyseal bone marrow, compare to their central marrow counterparts, have superior proliferative ability with reduced expression of cell cycle inhibitors[10]. Our results are in agreement with this finding and further suggest that nestin-positive cells in primary spongiosa of long bone proliferate much faster than nestin-negative cells in the same region during periods of rapid bone growth. Because these cells are likely no longer required in this particular region during adulthood, they stop proliferating and undergo senescence during late puberty. We also found that in early puberty, there are high percentage of Osx$^+$ osteoprogenitor cells in nestin-expressing cells in the primary spongiosa region, implicating a high proliferating property of these cells, likely because of a high demand for osteoblast replenishment for bone formation during this period of time. An interesting phenomenon from the present study is that while the number of the Nestin-GFP$^+$ cells gradually lost in the primary spongiosa of long bone during late puberty, LepR$^+$ cells gradually increased in this region. It was proposed, based on the recent fate-mapping studies[5,30,48], that there are two distinct types of bone stem/progenitor cells: growth-associated

MSPCs that reside in metaphyseal bone marrow to provide descendants in postnatal growing bone and peri-sinusoidal MSPCs such as LepR$^+$ cells as a major contributor to the remodeling process in adulthood. Our data suggests that nestin$^+$ cells in the primary spongiosa of long bones may represent growth-associated MSPCs, the function of which will be gradually replaced by adult stem/progenitor cells during late puberty.

Our work establishes the role of Ezh2-H3K27me3 as a key epigenetic regulator that controls the onset and progression of MSPC senescence during the transition of fast- to slow-growing phase of long bones. Ezh2-H3K27me3 modulates the senescence of MSPCs in a spatiotemporal manner. The self-renewal and proliferative capacity of cells in primary spongiosa of fast-growing bones are maintained by a high level of Ezh2-H3K27me3, whereas loss of Ezh2-H3K27me3 during late puberty leads to cell senescence. Moreover, in vivo study using genetic Ezh2 ablation mouse model revealed that removal of the H3K27me3 mark in nestin$^+$ cells in early pubertal mice resulted in premature cellular senescence, depleted MSPC pool, and resultant diminished osteogenesis. Consistent with the present finding, an in vivo work by Dudakovic et al.[49] reported that loss of Ezh2 in uncommitted mesenchymal cells (Prrx1-Cre) induces the expression of multiple CDK inhibitors (e.g., Cdkn1c/p57, Cdkn2a/p16), disrupted intra-membranous bone formation, and a reduction in trabecular bone volume in young mice. We did not detect any changes in the expression levels of Runx2 and Osteocalcin in the Ezh2-low Nestin-GFP$^-$ MSPCs vs. Ezh2-rich Nestin-GFP$^+$ MSPCs. The H3K27me3 mark on the promoters of Runx2 and Osteocalcin genes was also not affected, indicating that Ezh2-H3K27me3 does not directly regulate osteoblast differentiation of the cells. We did detect increased expression of p16$^{INK4a}$, p21$^{CIP1}$, p27$^{KIP1}$, and p15$^{INK4b}$ by reducing the H3K27me3 mark near the TSS on their promoter regions in the qRT-PCR and ChIP analysis. Therefore, the primary role of Ezh2 in MSPCs in primary spongiosa of long bone during childhood is to promote cell proliferation and prevent the cells from senescence to maintain the MSPC pool rather than to regulate their osteoblast commitment/differentiation. On the contrary, previous in vitro studies showed that Ezh2-H3K27me3 regulates the lineage differentiation of cultured bone marrow-derived MSPCs by repressing osteoblast commitment/differentiation genes[50–52], whereas inhibition of Ezh2-H3K27me3 in the cells promoted osteogenic differentiation. The different roles of Ezh2-H3K27me3 in regulating the behavior of the MSPCs in vitro and in vivo may be attributable to the different microenvironments. Stem/progenitor cell behavior are highly dependent on their microenvironment, and these cells coordinate niche signals (i.e., autocrine and paracrine factors) and chromatin states to choose appropriate fates[22]. Increasing evidence suggest that when the niche changes, cells dynamically change their chromatin landscape and remodel super-enhancers in response to the environmental change[53,54]. Another possible explanation for the discrepant results could be related to the different origins of the cells examined. In the present study, we primarily assessed the behavior of nestin-positive cells, which are neural crest-derived cells[55], mainly giving rise to vascular endothelial cells and osteoblasts in skeleton[27–29]. The mechanisms that epigenetically regulate the fate of this cell population could be different from mesoderm-derived MSPCs. The function of Ezh2 in regulating stem cell fate also highly correlates with developmental stage. For instance, although Ezh2 is essential for self-renewal and proliferation of epidermal progenitors[56] and blood progenitors[57] during embryonic development, it is not required for the proliferation of these cell types in adult mice.

Heterozygous missense mutations in Ezh2 cause Weaver syndrome, a rare condition characterized by tall stature, macrocephaly, accelerated osseous maturation[58,59]. A recent study

showed that H3K27 methylation regulates chondrocyte proliferation and hypertrophy in the growth plate, indicating that this particular epigenetic modification in chondrocytes regulates skeletal longitudinal growth[60]. Our finding that Ezh2-H3K27me3 regulates the cellular proliferation/senescence in primary spongiosa during skeletal growth/cessation implies another important role of H3K27 methylation in skeleton during this special period of life, i.e., regulating the rates of bone formation and bone mineral accrual to harmonize the speed of bone elongation.

Peak bone mass in childhood is not only a major determinant for the incidence of distal forearm fractures in this period, but also associated with bone mass and fracture risk later in life[4]. An epidemiologic study demonstrated that 60% of the risk of osteoporosis can be explained by the bone mineral acquired by early adulthood[4,61–63]. Our finding that deletion of *Ezh2* in nestin+ cells during early puberty increases the risk of osteoporosis in later adulthood suggests that premature cellular senescence in the primary spongiosa region during the prepubertal or early pubertal phase may also be a major cause of osteoporosis/bone loss in later life. Particularly, our finding that glucocorticoid induces premature senescence of MSPCs at primary spongiosa may represent a novel cellular mechanism for its deleterious skeletal effects during childhood. It is noteworthy to mention that the response of bone cells to glucocorticoids can be different between species[64,65]. Same dose of glucocorticoid used in mice in the current study may induce more severe (or does not induce) cellular senescence in humans. Future preclinical large animal models and/or clinical studies would be needed to determine whether the change is epigenetically controlled through the same mechanism.

## Methods

**Animals.** We purchased the *Nestin-Cre^ERT2* mice (stock no. 003771) and *Utx^flox/flox* mice (stock no. 024177) strain from the Jackson Laboratory (Bar Harbor, ME). *Nestin-GFP* mice were provided by Dr. Grigori Enikolopov at Cold Spring Harbor Laboratory (Cold Spring Harbor, NY). *Ezh2^flox/flox* mice (stock no. 15499) were obtained from Mutant Mouse Resource and Research Centers (University of North Carolina, Chapel Hill, NC).

*Nestin-Cre^ERT2* mice were crossed with *Ezh2^flox/flox* mice. The offspring were intercrossed to generate the *Nestin-Cre^ERT2; Ezh2^flox/flox* mice (*Ezh2* iKO) and *Nestin-Cre^ERT2* mice (*WT*). *Nestin-Cre^ERT2* mice were crossed with *Utx^flox/flox* mice. The offspring were intercrossed to generate the *Nestin-Cre^ERT2; Utx^flox/flox* mice (*Utx* iKO) and *Nestin-Cre^ERT2* mice (*WT*). To induce CreER activity, we injected mice at designed time points with tamoxifen (100 mg/kg B.W.) for different periods. The genotypes of the mice were determined by PCR analyses of genomic DNA extracted from mouse-tail snips using the following primers: *Nestin-Cre* forward, 5′-GCG GTC TGG CAG TAA AAA CTA TC-3′ and reverse, 5′-GTG AAA CAG CAT TGC TGT CAC TT-3′; *Nestin*-GFP allele forward, 5′-GGA GCT GCA CAC AAC CCA TTG CC-3′ and reverse, 5′-GAT CAC TCT CGG CAT GGA CGA GC-3′; loxP *Ezh2* allele forward, 5′-CTG CTC TGA ATG GCA ACT CC-3′ and reverse, 5′-TTA TTC ATA GAG CCA CCT GG-3′; loxP *Utx* allele forward, 5′-GGT CAC TTC AAC CTC TTA TTG GA-3′ and reverse, 5′-ACG AGT GAT TGG TCT AAT TTG G-3′.

To manipulate bone growth/accrual in puberty, 4-week-old *Nestin-GFP* mice were injected with growth hormone daily (5 mg/kg B.W. for 4 weeks). Four-week-old *Nestin-GFP* mice were injected with human PTH1-34 daily (80 μg/kg B.W. for 4 weeks). Two-week-old *Nestin-GFP* mice were injected with prednisolone daily at dosage of 10 mg/m$^2$/day for 2 weeks. Body surface area was calculated with Meeh's formula, using a $k$ constant of 9.82 for mice × body weight (g) to the two-thirds power (BSA = $kW^{2/3}$)[66]. Six-week-old *Nestin-GFP* mice were treated with GSK-J4 (100 mg/kg B.W.) daily for 2 weeks. Corresponding vehicle-treated mice were used as controls.

All animals were maintained in the animal facility of the Johns Hopkins University School of Medicine. The experimental protocol was reviewed and approved by the Institutional Animal Care and Use Committee of the Johns Hopkins University, Baltimore, MD, USA.

**Micro-CT analysis.** Mice femora were dissected, fixed overnight in 70% ethanol, and analyzed by high-resolution μCT (Skyscan 1172, Bruker MicroCT, Kontich, Belgium). We used NRecon image reconstruction software, version 1.6, (Bruker MicroCT), CTAn data-analysis software, version 1.9 (Bruker MicroCT), and CTVol three-dimensional model visualization software, version 2.0, (Bruker MicroCT) to analyze parameters of the trabecular bone in the metaphysis and cortical bone in the mid-diaphysis. The scanner was set at 50 kVp, 201 mA, and a resolution of 12.64 mm/pixel. Cross-sectional images of the distal femur were used to perform three-dimensional histomorphometric analysis of trabecular bone. The sample area selected for analyses was a 2-mm length of the metaphyseal secondary spongiosa, originating 1.0 mm below the epiphyseal growth plate and extending caudally. Cortical morphometry was analyzed within a 600-μm-long section at mid-diaphysis of the femur and included measurements of average thickness and cross-sectional area.

**Cell sorting and flow cytometry analysis.** For flow cytometric analysis and sorting of mouse CD45⁻GFP⁻ and CD45⁻GFP⁺ MSPCs from femoral metaphysis, we dissected the femora free of soft tissues from *Nestin-GFP* mice. The epiphysis was removed and only the 3-mm-long metaphysis regions were processed. The bone was then digested with a protease solution (2 mg/ml collagenase A and 2.5 mg/ml trypsin in phosphate-buffered saline [PBS]) for 20 min to remove the periosteum and periosteal progenitors (step I). The bones were cut into small pieces and digested in the protease solution for another 1 h (C, step II). Cells within the supernatant were collected for flow cytometry. After the process of red blood cell lysis with commercial ammonium-chloride-potassium lysis buffer (Quality Biological, Inc., Gaithersburg, MD), CD45⁺ cells were removed by CD45 MicroBeads using MACS cell separation system (Miltenyi Biotec, San Diego, CA). Cells were then sorted according to side scatter and GFP-FITC fluorescence at >10³ log Fl-1 (GFP-FITC) fluorescence after negative selection of leukocyte common antigen CD45 at <10³ log Fl-3 (CD45⁻PerCP) fluorescence. FACS was performed using a 5-laser BD FACS and FACSDiva (Becton Dickinson Biosciences, San Jose, CA). Flow cytometric analyses were performed using a FACSCalibur flow cytometer and CellQuest software (Becton Dickinson Biosciences). The primary antibodies used were FITC-conjugated anti-mouse GFP (Biolegend, 338008, 1:200), PerCP⁻conjugated anti-mouse CD45 (Biolegend, 103130, 1:200). For flow cytometric analysis of LepR⁺ cells, antibody against mouse leptin receptor (R&D Systems, AF497, 1:500) was used, followed by incubation with Cy3-conjugated secondary antibody (Jackson ImmunoResearch, 705-165-003, 1:200).

**Immunocytochemistry and immunofluorescence.** For immunocytochemical staining, we incubated cultured cells with primary antibody to mouse p16^INK4a (Abcam ab189034,1:200), Ezh2 (Cell Signaling 5246S,1:200) at 4 °C overnight and subsequently used a horseradish peroxidase–streptavidin detection system (DAKO) or Alexa Fluor-coupled secondary antibody (Jackson ImmunoResearch Laboratories, Inc., West Grove, PA) to detect immunoreactivity. For bromodeoxyuridine (BrdU) assay, cultured cells were incubated with BrdU solution (10 μM) at 37 °C overnight, followed by immunocytochemical staining with primary antibody against BrdU (Abcam, ab6326, 1:100). Senescent cells were detected by senescence β-Galactosidase staining kit according to the manufacturer's instructions (Cell Signaling Technology, Danvers, MA).

At the time of euthanasia, the femora were resected and fixed in 4% paraformaldehyde solution for 4 h, decalcified in 0.5 M EDTA at 4 °C with constant shaking for 1–2 days and immersed into 20% sucrose and 2% polyvinylpyrrolidone solution for 24 h. Finally, the tissues were embedded in optimal cutting temperature compound (Sakura Finetek USA, Inc., Torrance, CA) or 8% gelatin (porcine) in presence of 20% sucrose and 2% polyvinylpyrrolidone (PVP) solution[67]. Longitudinally oriented 10-μm-thick sections of bone, including the metaphysis and diaphysis, were processed for immunofluorescence staining.

For immunofluorescence staining, we incubated the sections with primary antibodies to mouse Ki67 (Novus Biologicals, NB500-170, 1:50), p16^INK4a (Abcam ab189034, 1:100), GFP (Rockland, 600-101-215, 1:500 or Abcam, ab290,1:200), Nestin (Aves Labs, NES, 1:100), Ezh2 (Cell Signaling 5246S, 1:200), H3K27me3 (Cell Signaling, 9733S, 1:800), osterix (Abcam, ab22552, 1:200), osteocalcin (Takara, M173, 1:200), endomucin (sc-65495, Santa Cruz, diluted 1:100), pecam1 (CD31) conjugated to Alexa Fluor 488 (R&D Systems, Inc., Minneapolis, MN, FAB3628G, 1:100), leptin receptor (R&D Systems, Inc., Minneapolis, MN, AF497, 1:200), followed by incubation with FITC, or Cy3-conjugated secondary antibodies (Jackson ImmunoResearch). Nuclei were counterstained with DAPI (Sigma). The sections were mounted with the ProLong Antifade Kit (Molecular Probes, Eugene, OR) and observed under a Fluo View 300 Confocal Microscope (Olympus America, Inc., Center Valley, PA).

**Chromatin immunoprecipitation and antibodies.** Chromatin immunoprecipitations were performed according to conditions suggested by the manufacturer's EpiTect ChIP OneDay kit (Qiagen, GA-101) with ChIP-grade antibodies to mouse H3K27me3 (Qiagen, GAM-9205, 1:50). Briefly, formaldehyde was added to cells to cross-link proteins to DNA, and the cells were lysed in 1.5 ml lysis buffer (50 mM HEPES, pH 7.5, 140 mM NaCl; 1 mM EDTA; 1% Triton X-100; 0.1% sodium deoxycholate; 0.1% sodium dodecyl sulfate). Cell lysates were sonicated at the set of 2 s on/15 s off for three rounds using a Bioruptor ultrasonic cell disruptor (Diagenode, Denville, NJ) to shear genomic DNA to an average fragment size of 150–250 bp. Of the sample, 1% was removed for use as an input control. ChIP was performed following protocol provided by EpiTect ChIP OneDay kit (Qiagen) using antibodies toward H3K27Me3 (Qiagen). Anti-RNA polymerase II and control IgG were used as positive and negative controls, respectively. After washing

and de-crosslinking, the precipitated DNA was purified using a QIAquick PCR purification kit (Qiagen).

**ChIP-qPCR**. ChIP-qPCR was performed using SYBR Green PCR Master Mix and 7900 HT Fast Real-Time PCR System (Applied Biosystems Corp., Foster City, CA). Primers for $p15^{INK4b}$, $p16^{INK4a}$, $p19^{ARF}$, $p21^{CIP1}$, $p27^{KIP1}$, Runx2, Osteocalcin, and Gapdh were used (see Supplementary Table 1 for primer sequences). Absolute quantification was performed and enrichment expressed as a fraction of the whole-cell extract control.

**Mouse epigenetic chromatin modification enzymes PCR array**. FACS-sorted $CD45^-GFP^+$ and $CD45^-GFP^-$ mesenchymal stem/progenitor cells in metaphysis of femora from 4-week-old Nestin-GFP mice were harvested, and total RNA was isolated using RNeasy Mini kit (Qiagen). RNA samples were assessed for quality and integrity using Synergy HT (Biotek Instruments, Inc., Winooski, VT). RNA was reverse transcribed to complementary DNA using SuperScript III First Strand Synthesis System for qRT-PCR (Invitrogen Corp., Carlsbad, CA) according to manufacturer's protocols. Customized mouse epigenetic chromatin modification enzymes $RT^2$ Profiler PCR array (Qiagen) was performed using SYBR Green PCR Master Mix and 7900 HT Fast Real-Time PCR System (Applied Biosystems Corp., Foster City, CA) and analyzed according to manufacturer's instructions. To validate the candidate genes screened from epigenetic chromatin modification enzymes PCR array, primers of Ezh2 and Ezh1 (Supplementary Table 2) were designed using primer bank database and Primer3, version 0.4.0, software (Whitehead Institute for Biomedical Research, Cambridge, MA). The amplification conditions were 95 °C for 10 min, 95 °C for 15 s, and 60 °C for 1 min. No-template and no-RT controls were included for each assay to ensure quality and complementary DNA specificity of the primers. Target-gene expression was normalized to glyceraldehyde 3-phosphate dehydrogenase (Gapdh) messenger RNA and relative gene expression assessed using the $2^{-\Delta\Delta CT}$ method.

**Statistics**. Data are presented as means ± standard errors of the mean. Unpaired, two-tailed Student's $t$-tests were used for comparisons between two groups. For multiple comparisons, one-way analysis of variance (ANOVA) with Bonferroni post hoc test was applied. All data were normally distributed and had similar variation between groups. Statistical analysis was performed using SAS version 9.3 software (SAS Institute, Inc., Cary, NC). $P < 0.05$ was deemed significant.

**Data availability**. The data that support the findings of this study are available within the article and Supplementary Files or available from the corresponding author upon reasonable request.

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

# ARTICLE

43. Xie, H. et al. PDGF-BB secreted by preosteoclasts induces angiogenesis during coupling with osteogenesis. *Nat. Med.* **20**, 1270–1278 (2014).
44. Kachinsky, A. M., Dominov, J. A. & Miller, J. B. Myogenesis and the intermediate filament protein, nestin. *Dev. Biol.* **165**, 216–228 (1994).
45. Delacour, A., Nepote, V., Trumpp, A. & Herrera, P. L. Nestin expression in pancreatic exocrine cell lineages. *Mech. Dev.* **121**, 3–14 (2004).
46. Sahlgren, C. M. et al. Cdk5 regulates the organization of Nestin and its association with p35. *Mol. Cell. Biol.* **23**, 5090–5106 (2003).
47. Park, D. et al. Nestin is required for the proper self-renewal of neural stem cells. *Stem Cells* **28**, 2162–2171 (2010).
48. Ono, N., Ono, W., Nagasawa, T. & Kronenberg, H. M. A subset of chondrogenic cells provides early mesenchymal progenitors in growing bones. *Nat. Cell Biol.* **16**, 1157–1167 (2014).
49. Dudakovic, A. et al. Epigenetic control of skeletal development by the histone methyltransferase Ezh2. *J. Biol. Chem.* **290**, 27604–27617 (2015).
50. Hemming, S. et al. EZH2 and KDM6A act as an epigenetic switch to regulate mesenchymal stem cell lineage specification. *Stem Cells* **32**, 802–815 (2014).
51. Ye, L. et al. Histone demethylases KDM4B and KDM6B promotes osteogenic differentiation of human MSCs. *Cell Stem Cell* **11**, 50–61 (2012).
52. Hemming, S. et al. Identification of Novel EZH2 targets regulating osteogenic differentiation in mesenchymal stem cells. *Stem Cells Dev.* **25**, 909–921 (2016).
53. Gosselin, D. et al. Environment drives selection and function of enhancers controlling tissue-specific macrophage identities. *Cell* **159**, 1327–1340 (2014).
54. Lavin, Y. et al. Tissue-resident macrophage enhancer landscapes are shaped by the local microenvironment. *Cell* **159**, 1312–1326 (2014).
55. Isern, J. et al. The neural crest is a source of mesenchymal stem cells with specialized hematopoietic stem cell niche function. *Elife* **3**, e03696 (2014).
56. Ezhkova, E. et al. Ezh2 orchestrates gene expression for the stepwise differentiation of tissue-specific stem cells. *Cell* **136**, 1122–1135 (2009).
57. Mochizuki-Kashio, M. et al. Dependency on the polycomb gene Ezh2 distinguishes fetal from adult hematopoietic stem cells. *Blood* **118**, 6553–6561 (2011).
58. Gibson, W. T. et al. Mutations in EZH2 cause Weaver syndrome. *Am. J. Hum. Genet.* **90**, 110 (2012).
59. Tatton-Brown, K. et al. Germline mutations in the oncogene EZH2 cause Weaver syndrome and increased human height. *Oncotarget* **2**, 1127–1133 (2011).
60. Lui, J. C. et al. EZH1 and EZH2 promote skeletal growth by repressing inhibitors of chondrocyte proliferation and hypertrophy. *Nat. Commun.* **7**, 13685 (2016).
61. Cooper, C. & Melton, L. J. Epidemiology of osteoporosis. *Trends Endocrinol. Metab.* **3**, 224–229 (1992).
62. Rosen, C. J. *The Epidemiology and Pathogenesis of Osteoporosis* (2000).
63. Veldhuis-Vlug, A. G. & Rosen, C. J. Mechanisms of marrow adiposity and its implications for skeletal health. *Metabolism* **67**, 106–114 (2016).
64. Thiele, S., Baschant, U., Rauch, A. & Rauner, M. Instructions for producing a mouse model of glucocorticoid-induced osteoporosis. *Bonekey Rep.* **3**, 552 (2014).
65. Ersek, A. et al. Strain dependent differences in glucocorticoid-induced bone loss between C57BL/6J and CD-1 mice. *Sci. Rep.* **6**, 36513 (2016).
66. Cheung, M. C. et al. Body surface area prediction in normal, hypermuscular, and obese mice. *J. Surg. Res.* **153**, 326–331 (2008).
67. Kusumbe, A. P., Ramasamy, S. K., Starsichova, A. & Adams, R. H. Sample preparation for high-resolution 3D confocal imaging of mouse skeletal tissue. *Nat. Protoc.* **10**, 1904–1914 (2015).

## Acknowledgements
The authors gratefully acknowledge the assistance of Eileen Martin and Rachel Box at the Johns Hopkins Department of Orthopaedic Surgery Editorial Services in editing the manuscript. This work was supported by the National Institutes of Health DK083350 (to M.W.).

## Author contributions
M.W. and C.L. designed the experiments; C.L. and Y.C. carried out most of the experiments; L.W., B.G. and H.C. helped to collect the samples. X.C., J.L.C., P.G., F.-Q.Z., X.L. and B.Y. proofread the manuscript; M.W. supervised the experiments, analyzed results and wrote the manuscript.

## Additional information

**Competing interests:** The authors declare no competing financial interests.

