## [Peer Review file · Nature Communications]

Reviewers' comments:

Reviewer #1 (Remarks to the Author):

This manuscript contains an impressive amount of data that are interpreted to support some interesting conclusions. The conclusions about cellular senescence in the growth plate late in puberty are well substantiated and important. The conclusions about the epigenetic changes that drive the process are based on more tenuous evidence and contradict observations by others. However the authors have pursued the experiments with state-of-the-art techniques, and results are likely to stimulate other experiments in the field.

The data make a very convincing story that growth ceases late in puberty because mesenchymal stem/progenitor cells in the primary spongiosa layer of long bones undergo senescence, as senescence is carefully defined here. The authors support this conclusion with the expected changes in SA-Gal+, K67+, BrdU+, p16INK4A+, p21+ and Nestin+ cells (Figs.1-2). The results persuade them to focus on Nestin+ as the cells that are lost with senescence and growth arrest. And they observe the expected effects with treatment with GH, iPTH, and prednisolone (Fig.3). They then attempt to identify epigenetic changes that produce these effects. Here they follow the kind of logic currently in vogue to select such targets: "Profiling the expression of 86 key genes encoding 4 enzymes that modify genomic DNA and histones showed 12 differentially expressed genes in these 2 cell populations (1 downregulated and 11 upregulated; Fig. 4A)." This is of course a major step of selection with a limited rationale, since it arbitrarily limits the number of genes examined. At any rate it leads them to the histone methyltransferase Ezh2 that specifically catalyzes histone H3K27me3 and is the only such enzyme down regulated in a comparison of Nestin positive versus negative cells from the metaphysis of bone. They then indicate that H3K27me3 was "enriched in the promoter regions of the cell senescence inducer genes p16INK4a and p21CIP1". And smaller changes in other senescence related genes. But there is no indication of how much of the promoter regions were examined, or of other regions of the genes likely to contain enhancers. At any rate, they proceed to produce a conditional knock out mouse for Ezh2 (Nestin-CreERT2::Ezh2 flox/flox). The knock out is only partially effective (level reduced from 70% positive cells to 30%; Fig. 4E). However, they observed the expected effects on Nestin+ cells and bone growth (Fig. 5). They then compliment these data with data on the effects of a demethylase inhibitor (GSK-J4 against activity of JMJD3 and UTX) with or without prednisolone (Fig. 6). The inhibitor was referred to as specific in the mice although there are data on dose or specificity in the paper or the reference cited.

In effect there are several questionable steps in generating the epigenetic data. A saving grace is that the same Ezh2 pathway has been found to be important in many other systems. The only problem is, as the authors indicate, the results obtained by others are the opposite of the results here: decreased expression of Ezh2-H3K27me3 has been shown to increase osteogenesis and not decrease it.

Suggestions:

1. The Discussion should be re-written to give a more sophisticated account of why they observed epigenetic changes the opposite of others, e.g. limitations of present techniques and knowledge of epigenetics, in vivo vs. in vitro studies, autocrine and paracrine factors that might trigger the epigenetic changes, the limitations of their data (selection of genes, assay of histones in promoter regions only, partial knock down of Ezh2, etc.).
2. The authors should provide more data on the histone modification (H3K27me3) they assayed in promoter regions of genes. Also whether any control genes were assayed.
3. Data with the inhibitor GSK-J4 should be eliminated unless the authors can show it was specific and non-toxic in vivo at the dose used in the mice.

4. Data are incomplete in several figures. P values not indicated for some of the FACS graphs with small changes (Fig. 3C and 4O). Some abbreviations not explained (Fig. 1 C). Some graphs not adequately labeled (Fig. 6 J and K).

Reviewer #2 (Remarks to the Author):

This study examines the role of PRC2 and UTX related methylation of H3K27 contributing cellular senescence of in mesenchymal progenitor cells (MPC) during puberty. Using several specific transgenic and conditional knockout mouse models the authors demonstrate during late puberty, there is a reduction in nestin-positive MPCs, cellular proliferation and a concurrent increase in cellular senescence in the primary spongiosa. The authors further demonstrate that recruitment of Nestin-positive MPCs are increased by administration of GH or PTH and are significantly reduced in mice treated with glucocorticoids (prednisolone). Using a qPCR candidate approach, the authors identified Ezh2 as being down-regulated in CD45-GFP- MPCs relative to CD45-GFP+. Relatedly, the promoter regions of p16INK4a and p21CIP1 were enriched in H3K27me3 in the Nestin-GFP+ MPCs, reflecting the down-regulation of Ezh2 in these cells. The authors then demonstrated that in Nestin-CreERT2::Ezh2 flox/flox (Ezh2 iKO) mouse model there was a significant increase in SA- β Gal+ cells in the primary spongiosa. In addition, the authors demonstrate that by using a pharmacological inhibitor of UTX/JMJ3 increased the number of progenitor cells in the primary spongiosa and reversed the bone loss observed in mice treated with glucocorticoid during early puberty.

The authors should be commended on their well-conducted study and engaging manuscript. There are no major criticisms of the experimental design and all the experiments appear to be properly controlled, appropriately described and interpreted. As very minor points there should be some consideration given to the differences in responses to glucocorticoids in mice versus humans, specifically in bone and how these specific differences may temper these findings. These concerns could be addressed in the discussion (last paragraph).

Specific comments

[PG10LN15] Syntax: "To examined whether ..."

Point-by-point response to reviewers' comments

We would like to thank both reviewers for their positive, insightful, and constructive comments regarding our manuscript. We have addressed all of the questions and concerns brought forth through additional experimentation and clarification. The following responses have been prepared to address all of the reviewers' comments in a point-by-point fashion.

Response to comments from Reviewer #1:

General Comments:

This manuscript contains an impressive amount of data that are interpreted to support some interesting conclusions. The conclusions about cellular senescence in the growth plate late in puberty are well substantiated and important. The conclusions about the epigenetic changes that drive the process are based on more tenuous evidence and contradict observations by others. However the authors have pursued the experiments with state-of-the-art techniques, and results are likely to stimulate other experiments in the field.

The data make a very convincing story that growth ceases late in puberty because mesenchymal stem/progenitor cells in the primary spongiosa layer of long bones undergo senescence, as senescence is carefully defined here. The authors support this conclusion with the expected changes in SA-Gal+, K67+, BrdU+, p16INK4A+, p21+ and Nestin+ cells (Figs.1-2). The results persuade them to focus on Nestin+ as the cells that are lost with senescence and growth arrest. And they observe the expected effects with treatment with GH, iPTH, and prednisolone (Fig.3).

They then attempt to identify epigenetic changes that produce these effects. Here they follow the kind of logic currently in vogue to select such targets: "Profiling the expression of 86 key genes encoding 4 enzymes that modify genomic DNA and histones showed 12 differentially expressed genes in these 2 cell populations (1 downregulated and 11 upregulated; Fig. 4A)." This is of course a major step of selection with a limited rationale, since it arbitrarily limits the number of genes examined. At any rate it leads them to the histone methyltransferase Ezh2 that specifically catalyzes histone H3K27me3 and is the only such enzyme down regulated in a comparison of Nestin positive versus negative cells from the metaphysis of bone. They then indicate that H3K27me3 was "enriched in the promoter regions of the cell senescence inducer genes p16INK4a and p21CIP1". And smaller changes in other senescence related genes. But there is no indication of how much of the promoter regions were examined, or of other regions of the genes likely to contain enhancers. At any rate, they proceed to produce a conditional knock out mouse for Ezh2 (Nestin-CreERT2::Ezh2 flox/flox). The knock out is only partially effective (level reduced from 70% positive cells to 30%; Fig. 4E). However, they observed the expected effects on Nestin+ cells and bone growth (Fig. 5). They then compliment these data with data on the effects of a demethylase inhibitor (GSK-J4 against activity of JMJD3 and UTX) with or without prednisolone (Fig. 6). The inhibitor was referred to as specific in the mice although there are data on dose or specificity in the paper or the reference cited.

In effect there are several questionable steps in generating the epigenetic data. A saving grace is that the same Ezh2 pathway has been found to be important in many other systems. The only problem is, as the authors indicate, the results obtained by others are the opposite of the results here: decreased expression of Ezh2-H3K27me3 has been shown to increase osteogenesis and not decrease it.

Response: We thank the reviewer for his/her careful reading and positive overview of the manuscript. We also appreciate the reviewer's constructive suggestions, which were of great help in revising the manuscript.

To address the concern on the selection of the target senescence genes and their promoter regions in our analysis of H3K27me3 modification, we have performed additional ChIP assays and assessed the changes in H3K27me3 status at different promoter regions of the key genes involved in cell senescence (Fig. 4C–4G). Detailed information on how the promoter regions of the genes were selected has been added in the Methods section. The new results have been described and discussed in the Results and the Discussion section, respectively. Please see below the detailed response to Specific Concern #2.

We apologize for not clearly describing the deletion efficiencies in the inducible *Ezh2* KO mice used in our experiments. In fact, to achieve different purposes, we used different strategies for Tamoxifen administration and calculation of the deletion efficiency in different experiments. In **Fig. 5**, in order to demonstrate the negative correlation of the expressions of *Ezh2* and SA- β Gal (a senescence marker) in MSPCs in metaphysis of long bone, we injected a single dose of tamoxifen into the mice. As a result, *Ezh2*⁺ cells was reduced from 75% to <30% in the isolated MSPCs (Fig. 5C), which are a heterogeneous population containing both nestin⁺ and nestin⁻ cells. More importantly, we found that the expressions of *Ezh2* and SA- β Gal were mutually exclusive in the cells (Fig. 5B and 5C). In **Fig. 6**, to test whether deletion of *Ezh2* in nestin⁺ cells causes changes in blood vessels and osteogenesis, we injected three doses of tamoxifen into the mice. *Ezh2*-expressing nestin⁺ cells were reduced from 92% to approximately 16% (Fig. 6C). After *Ezh2* was deleted in the majority of nestin⁺ cells in primary spongiosa, osteogenesis and bone formation were significantly decreased. In the revised manuscript, we have added schematic diagrams (Fig. 5A, 5E, 6A, and 7A) to show the study design and different time courses of the inducible *Ezh2* KO mice in different experiments. Detailed descriptions on the experiments have also been added in the corresponding Figure Legends and the Results section.

We agree with the reviewer that the effect of the demethylase inhibitor GSK-J4 is not bone-specific. In the revised manuscript, we have eliminated the data showing the effect of GSK-J4 on attenuating glucocorticoid-induced bone loss (i.e. original Fig. 6).

Regarding the discrepant results obtained from the present study and previous *in vitro* studies by others on the role of *Ezh2*-H3K27me3 in regulating osteogenesis, we have fully discussed the possible reasons (page 14, lines 16–34; page 15, lines 1–21). In addition, we have provided new data showing that *Ezh2*-H3K27me3 does not directly regulate the master osteoblast differentiation inducer genes in MSPCs in our model system (new Fig. 4G, 4L, and 4M). Please also see below the detailed response to Specific Concern #1.

Specific Comments:

1. *The Discussion should be re-written to give a more sophisticated account of why they observed epigenetic changes the opposite of others, e.g. limitations of present techniques and knowledge of epigenetics, in vivo vs. in vitro studies, autocrine and paracrine factors that might trigger the epigenetic changes, the limitations of their data (selection of genes, assay of histones in promoter regions only, partial knock down of Ezh2, etc.).*

Response: As the reviewer pointed out, our *in vivo* study demonstrates that deletion of *Ezh2* in nestin⁺ cells resulted in cellular senescence, depleted mesenchymal stem/progenitor cell pool,

and reduced osteogenesis. However, previous *in vitro* studies showed that inhibition of Ezh2-H3K27me3 in the cultured bone marrow-derived MSPCs promoted osteogenic differentiation of the cells. In the revised manuscript, possible reasons for the discrepant results have been fully discussed, such as difference in *in vitro* and *in vivo* studies, different origins of the cells and their responses to microenvironment. The corresponding paragraph has been fully re-written (page 14, lines 16–34; page 15, lines 1–21). In addition, we have performed new experiments to test whether Ezh2-H3K27me3 directly regulates osteoblast differentiation of MSPCs in our model system. The changes in gene expression and H3K27me3 status at the promoter regions of master osteoblast differentiation genes Runx2 and osteocalcin have been assessed by qRT-PCR and ChIP-qPCR assays. We could not detect any changes in the expression levels of Runx2 and osteocalcin in the Ezh2-low Nestin-GFP⁻ MSPCs vs. Ezh2-rich Nestin-GFP⁺ MSPCs (new Fig. 4L and 4M). Consistently, H3K27me3 mark on the promoters of Runx2 and osteocalcin was not detected (new Fig. 4G), indicating that Ezh2-H3K27me3 does not directly regulate osteoblast differentiation of MSPCs in our model system.

2. *The authors should provide more data on the histone modification (H3K27me3) they assayed in promoter regions of genes. Also whether any control genes were assayed.*

Response: In the past two months, we have performed additional ChIP-qPCR assays and assessed the changes in histone methylation status at different promoter regions of the key genes involved in cell senescence. Specifically, the changes in H3K27me3 at the INK4b-ARF-INK4a locus, which encodes INK4 family member p15INK4b and p16INK4a, and a tumor suppressor p19/p14 ARF, have been detected. Please note that we chose 3 promoter regions for each gene (regions just upstream, surrounding, and within the intron just downstream of the transcription start site) in the ChIP-qPCR assays. Changes in H3K27me3 at the promoter regions of other cell senescence-related genes, including cell cycle inhibitor genes p21CIP1 and p27KIP1, have also been detected. We have also now assessed the changes in H3K27me3 at the promoter regions of master osteoblast differentiation inducer genes Runx2 and osteocalcin. Please note that we chose the same promoter regions of the genes that were used in the *in vitro* studies reported previously (*Cell Stem Cell* 2012; 11: 50–61; *Stem Cells* 2014; 32:802–815; *FASEB J* 2017; 31:1011–1027). In each of these assays, an IgG antibody has been used as a negative IP control, and the promoter region of GAPDH has been used as a negative qPCR control. The results obtained from these new experiments suggest that the primary role of Ezh2 in MSPCs in primary spongiosa of long bone during childhood is to promote cell proliferation and prevent the cells from senescence to maintain the MSPC pool rather than to regulate their osteoblast commitment/differentiation.

In the revised manuscript, detailed information on how the promoter regions of the genes were selected has been described in the Methods section (page 19, lines 32–33; Supplementary Table 1). The new data obtained from the experiments have been included in Fig. 4C–4G. Description and discussion regarding the results and the conclusion drawn from the finding have been added in the Results (page 9, lines 5–34; page 10, lines 1–8) and the Discussion section (page 14, lines 16–34; page 15, lines 1–21).

3. *Data with the inhibitor GSK-J4 should be eliminated unless the authors can show it was specific and non-toxic in vivo at the dose used in the mice*

Response: As the reviewer suggested, we have removed the data with the inhibitor GSK-J4 (original Fig. 6).

4. *Data are incomplete in several figures. P values not indicated for some of the FACS graphs with small changes (Fig. 3C and 4O). Some abbreviations not explained (Fig. 1 C). Some graphs not adequately labeled (Fig. 6 J and K).*

Response: We are sorry for the unclear labeling and insufficient description of the figures in the original submission. We have carefully edited the manuscript to correct the mentioned errors and others not specifically mentioned. Fig. 3C is one of the representative flow cytometry images showing the percentage of the CD45⁻GFP⁺ cells isolated from femoral metaphysis in 1 mouse, and Fig. 3D shows the statistical calculation of the CD45⁻GFP⁺ cells from 5 mice (per treatment group) with *p* value presented. Similarly, new Fig. 5O is one of the representative flow cytometry images showing the percentage of the CD45⁻GFP⁺ cells isolated from femoral metaphysis in 1 mouse, and new Fig. 5P shows the statistical calculation of the CD45⁻GFP⁺ cells from 5 mice (per treatment group) with *p* value presented. In the revised manuscript, more detailed descriptions of these figures have been added in the figure legends. Abbreviations have been added in all legends. The original Fig. 6 has been removed from the revised manuscript as discussed above.

Response to comments from Reviewer #2:

General Comments:

This study examines the role of PRC2 and UTX related methylation of H3K27 contributing cellular senescence of in mesenchymal progenitor cells (MPC) during puberty. Using several specific transgenic and conditional knockout mouse models the authors demonstrate during late puberty, there is a reduction in nestin-positive MPCs, cellular proliferation and a concurrent increase in cellular senescence in the primary spongiosa. The authors further demonstrate that recruitment of Nestin-positive MPCs are increased by administration of GH or PTH and are significantly reduced in mice treated with glucocorticoids (prednisolone). Using a qPCR candidate approach, the authors identified Ezh2 as being down-regulated in CD45-GFP⁻ MPCs relative to CD45-GFP⁺. Relatedly, the promoter regions of p16INK4a and p21CIP1 were enriched in H3K27me3 in the Nestin-GFP⁺ MPCs, reflecting the down-regulation of Ezh2 in these cells. The authors then demonstrated that in Nestin-CreERT2::Ezh2 flox/flox (Ezh2 iKO) mouse model there was a significant increase in SA-βGal⁺ cells in the primary spongiosa. In addition, the authors demonstrate that by using a pharmacological inhibitor of UTX/JMJ3 increased the number of progenitor cells in the primary spongiosa and reversed the bone loss observed in mice treated with glucocorticoid during early puberty.

The authors should be commended on their well-conducted study and engaging manuscript. There are no major criticisms of the experimental design and all the experiments appear to be properly controlled, appropriately described and interpreted. As very minor points there should be some consideration given to the differences in responses to glucocorticoids in mice versus humans, specifically in bone and how these specific differences may temper these findings. These concerns could be addressed in the discussion (last paragraph).

Response: We sincerely thank the reviewer for his/her in-depth review of this manuscript, overall encouragement, and valuable suggestion. In the revised manuscript, different responses of bone cells to glucocorticoids in mice versus humans, as well as the potential impact of the differences on our findings, have been discussed (page 16, lines 8–19).

Specific Comments

[PG10LN15] Syntax: “To examined whether ...”

Response: We apologize for the error. We have thoroughly checked the manuscript and corrected the typographical errors.

REVIEWERS' COMMENTS:

Reviewer #1 (Remarks to the Author):

The authors have responded as adequately as possible to the original criticisms.

Reviewer #2 (Remarks to the Author):

Accept

Point-by-point response to reviewers' comments

Again, we sincerely thank both reviewers for their big efforts in reviewing our manuscript. We appreciate the reviewer's valuable comments and constructive suggestions, which have helped to significantly improve the manuscript. The following responses have been prepared to address the reviewers' comments in a point-by-point fashion.

Response to comments from Reviewer #1:

Reviewer #1 (Remarks to the Author):

The authors have responded as adequately as possible to the original criticisms.

Response: We are pleased that the reviewer is satisfied with our responses to his/her critiques.

Response to comments from Reviewer #2:

Reviewer #2 (Remarks to the Author):

Accept

Response: We are delighted that the reviewer is satisfied with our revised manuscript.